# Assessing Safety Risks and Quantization-aware Safety Patching for Quantized Large Language Models

## ABSTRACT

Quantized large language models (LLMs) have garnered surging demand for broadening the deployment scenarios of LLMs, particularly on resource-constrained applications, which would otherwise be infeasible due to the substantial resource overhead incurred by astronomical model sizes. Propelled by this vast application potential, various quantization techniques have been developed to convert high-precision LLMs into low-precision quantized counterparts, aiming to preserve strong capabilities with reduced bit-widths. While these techniques have made significant strides in preserving utility, their implications for safety remain insufficiently studied. Recent findings highlight the fragility of safety mechanisms in both high-precision and quantized LLMs, underscoring the need for systematic safety evaluations and targeted interventions for quantized models.

In this paper, we present a comprehensive safety evaluation of quantized LLMs to complement existing efforts, covering four mainstream quantization techniques across diverse settings, including varying quantization bit-widths and different quantization-assisting datasets, through widely-accepted safety measurements. Our empirical evaluation reveals concerning safety degradation across all quantization methods and settings. To address this, we propose a quantization-aware safety patching framework, `Q-resafe`, to efficiently restore the safety capabilities of quantized LLMs while minimizing any adverse impact on utility. Extensive experiments demonstrate that `Q-resafe` effectively restores the safety of quantized LLMs obtained from diverse quantization processes, aligning closely with pre-quantization LLMs, even when evaluated against challenging datasets. We will make our implementation publicly available https://anonymous.4open.science/r/Qresafe-D085/.

## 1 INTRODUCTION

Large language models (LLMs)(Touvron et al., 2023; Anil et al., 2023; Achiam et al., 2023) continue to gain increasing applications across a wide spectrum of areas, offering astounding performance that often surpasses human capabilities in tasks ranging from general language processing and reasoning (Reizinger et al., 2024; Almeida et al., 2024) to more intricate and specialized domains such as medical assistance, education, autonomous vehicles, law, and finance (Ghosh et al., 2024; cop, 2023; He et al., 2024). Underpinning such surging demand and remarkable capabilities is the colossal model size (Huang et al., 2024), which however poses significant challenges for deploying LLMs on commodity and edge devices due to the overwhelming resource overhead in terms of memory footprint, computational cost, and energy consumption (Frantar et al., 2022; Xiao et al., 2023). Consequently, this has led to the growing popularity and importance of quantization on LLMs (Frantar & Alistarh, 2023), a primary technique for converting the original LLMs from the high-precision representation (e.g., 16-bit) to low-precision representation with reduced bit-widths, such as 8-bit, 4-bit, or even 1-bit (Kim et al., 2024a; Ma et al., 2024). Quantization on LLMs is desirable and sometimes even essential across various deployment scenarios. These include edge computing for real-time applications like autonomous vehicles, where delays are intolerable for interacting with resource-abundant cloud servers (Lin et al., 2023); Data security-critical scenarios that mandate keeping inference data on local commodity computing devices; Multi-tenant serving scenarios to reduce the storage overhead of multiple adaptations of LLMs for cloud service providers Chen et al. (2024).

**Safety capability of quantized LLM studies.** Both academia (Huang et al., 2024) and industry (cha, 2023; bin, 2023) have reached the consensus that merely chasing high utility is insufficient for the reliable adoption of LLMs. Safety capabilities are indispensable, in order to prevent harmful behaviors (Qi et al., 2024) such as generating content involving discrimination, spreading misinformation, or violating human values and social norms. Recent studies on high-precision LLMs find that safety is fragile to maintain, as even well-aligned LLMs can experience degraded safety alignment after slight fine-tuning (Li et al., 2023a) and become more vulnerable to be compromised by jailbreak examples (cwe, 2023; Li et al., 2024b). While these vulnerabilities are concerning for full-precision models, quantization processes exacerbate these risks by altering the weights of well-aligned models, often greater to extent than slight finetuning.

Consequently, understanding and preserving the safety capabilities of quantized LLMs is arguably even more crucial than for their full-precision counterparts, which are often managed by professional service providers. For instance, in on-device deployment scenarios of quantized LLMs, users typically lack the technical expertise to make informed decisions when jailbreaks occur, and edge devices lack the resources to implement the safety alignment of their models. Prior work has explored the safety aspects of quantized LLMs from various perspectives and revealed that quantized LLMs indeed suffer from degraded safety capabilities (Belkhiter et al., 2024; Egashira et al., 2024; Hong et al., 2024; Pan et al., 2021). Complementing existing safety studies on LLMs, there raises important research questions: *To what extent do different quantization techniques degrade the safety capabilities of quantized LLMs, and how can such declines in safety capabilities be mitigated?*

**Our work.** In this paper, we perform a systematic safety risk assessment of quantization on LLMs design to complement existing studies and mitigate the safety degradation by proposing a novel **Q**uantization-awa**re safe**ty patching algorithm (`Q-resafe`) to re-align the safety performance of quantized LLMs with their pre-quantization counterparts.

Safety risks assessment: Our assessment covers all four mainstream categories of LLM quantization techniques covering two post-quantization techniques and two quantization-aware training/finetuning techniques. To ensure the evaluated methods are sufficiently representative within each category, the selection criteria are based on whether the method is a seminal work with high citations (Lin et al., 2023; Liu et al., 2023b; Dettmers et al., 2024) or achieves state-of-the-art performance (Egiazarian et al., 2024), as detailed in Section 3.1. For quantization techniques that require an additional quantization-assisting dataset, we consider three datasets with varying safety risk levels: a directly harmful dataset, an indirectly harmful dataset, and a benign dataset. In addition, we evaluate quantized LLMs with two commonly adopted bit-widths. For safety risk measurement, we follow the well-established practice for full-precision LLMs (Li et al., 2023a) to ensure comprehensiveness. Our safety assessment results reveal that all four categories of quantization techniques lead to degraded safety capabilities. In general, post-quantization methods result in greater safety decline when compared to the quantization-aware finetuning methods with benign quantization-assisting datasets (calibration datasets or finetuning datasts depending on the specific quantization technique). This is because, given the same bit-width, post-quantization is inferior to quantization-aware finetuning in preserving the overall capabilities of LLMs, including both utility and safety. Quantized LLMs with higher bit-width (e.g., INT8) in general exhibit better safety capabilities compared to those with lower bit-width (e.g., INT4). Quantization-aware fine-tuning methods with benign datasets still incur safety declines because their objective centers on preserving utility, often neglecting safety-specific consideration. For instance, their finetuning datasets are utility-centered, and the objective function focuses on maintaining perplexity or downstream accuracy. Moreover, quantization-aware finetuning methods suffer a dramatic drop in safety if the quantization-assisting datasets contain harmful samples, suggesting that these datasets should be carefully scrutinized.

Safety risk patching: Propelled by the safety concern of quantized LLMs exposed by our assessment, we propose the first safety-patching framework, namely `Q-resafe`, tailor-made to restore the safety of quantized LLMs. Based on the evaluations, quantized LLMs generally exhibit satisfactory utility, as the quantized weights are carefully generated by existing quantization methods through a utility-centered design. Moreover, `Q-resafe` exploits DPO Rafailov et al. (2024), a popular technique for LLM alignment, as the loss function and proposes to construct a safety-patching dataset under the guidance of pre-quantization LLMs, which serves the purpose of transferring the safety capabilities to the quantized LLM during safety-patching.

The main contributions of this paper can be summarized as follows:

- We present a comprehensive safety evaluation of quantized LLMs to complement existing studies, covering four different quantization techniques and revealing significant safety implications;
- We propose `Q-resafe`, an efficient algorithm designed to mitigate the identified safety risks in quantized LLMs;
- We conduct extensive experiments to demonstrate the effectiveness of `Q-resafe` in restoring the safety capabilities for quantized LLMs.

## 2 RELATED WORKS

### 2.1 QUANTIZATION ON LLMs

Quantization is a model compression technique that reduces the storage requirements of a model by mapping high-precision values to low-precision values. Existing methods can be roughly divided into Post-training quantization(**PTQ**) (Frantar et al., 2022; Cheng et al., 2023; Xiao et al., 2023; Dettmers et al., 2023; Lee et al., 2023; Kim et al., 2023; Li et al., 2024a; Yao et al., 2022; Wei et al., 2022; 2023; Yuan et al., 2023; Lin et al., 2023; Liu et al., 2023a; Ashkboos et al., 2023; Li et al., 2023b; Ashkboos et al., 2024; Kim et al., 2024b; Shao et al., 2023; Zhao et al., 2024) and Quantization-aware training (**QAT**). In general, PTQs tend to be less effective than QAT, because QAT integrates the quantization into the training and helps the model adapt to lower accuracy, thus improving performance. But quantization-aware with full-parameter finetuning (Liu et al., 2023b; Du et al., 2024; Ma et al., 2024; Xu et al., 2024a) is heavily dependent on the data itself and requires more training effort, so it is currently not as widely explored in LLMs. Therefore, parameter-efficient finetuning (**PEFT**) (Li et al., 2023e; Guo et al., 2023; Xu et al., 2023; Chai et al., 2023; Hayou et al., 2024; Kim et al., 2024a; Dettmers et al., 2024) is introduced with the aim of creating models with high accuracy and low computational overhead.

### 2.2 SAFETY EVALUATIONS FOR LLMs

Safety in LLMs refers to their ability to avoid generating harmful, biased, or false information, ensuring that they behave in a compliant, helpful, honest, and harmless manner (cha, 2023). Exploring and evaluating the safety of LLMs is crucial because these models are increasingly deployed in real-world applications where they can inadvertently propagate toxic or misleading outputs. Safety is typically evaluated by testing whether LLMs follow harmful instructions, generate prohibited content, or display biases (Zou et al., 2023; Shi et al., 2024). Safety aspects have been extensively studies in full-precision LLMs (Zhan et al., 2023; Qi et al., 2023; Shayegani et al., 2023), systematically covering aspects like bias, toxicity, and robustness to adversarial attacks.

### 2.3 SAFETY EVALUATIONS FOR QUANTIZED LLMs

Very recently, several studies have pioneered the exploration of safety issues in quantized LLMs from various perspectives. For instance, Egashira et al. (2024) investigates safety vulnerabilities in quantized models and proposes a three-stage attack framework. Belkhiter et al. (2024) studies the robustness of AWQ and GPTQ techniques on Vicuna and developed benchmark datasets for harm-level evaluation. (Kumar et al., 2024b; Hong et al., 2024) analyzed different compression techniques across multiple LLMs, examining their impact on model safety and utility. In addition, Pan et al. (2021) revealed security risks in third-party quantized neural networks, where backdoor attacks can remain dormant in full-precision models but activate through quantization.

### 2.4 ALIGNMENT METHODS FOR LLMs

Traditional alignment techniques for full-precision LLMs such as instruction tuning (Peng et al., 2023), reinforcement learning from human feedback (RLHF) (Christiano et al., 2017; Ouyang et al., 2022; Bai et al., 2022), and direct preference optimization(DPO) (Rafailov et al., 2024) are widely used to align pre-trained models with human preference. These methods help models improve their outputs through explanations or justifications, which can serve as additional supervision signals. While LLMs can be trained to refuse inappropriate queries in many scenarios, ensuring consistently

safe output generation remains challenging. For example, Zephyr explores preference optimization by distilling feedback from multiple AI evaluators into a more efficient self-supervised process (Song et al., 2024; Wang et al., 2024; Tunstall et al., 2023).

While alignment methods for full-precision LLMs continue to develop, research on safety alignment approaches for quantized LLMs remains limited (Badshah & Sajjad, 2024; Xu et al., 2024b; Paglieri et al., 2024). Quantization modifies the model's internal representations, potentially affecting its adherence to safety and ethical guidelines established during full-precision training (Trukhanov & Soloveychik, 2024; Huang et al., 2024; Hu et al., 2024). Developing effective methods to maintain or enhance safety capabilities in quantized LLMs while preserving their efficiency benefits represents an important research direction.

# 3 ASSESSING SAFETY RISKS OF QUANTIZATIONS ON LLMS

## 3.1 SETUP OF ASSESSMENT

**Quantization Methods.** We cover all four mainstream categories of quantization techniques for a systematic evaluation of safety risks. In particular, we assess four prominent quantization methods from each category: AWQ, AQLM, LLM-QAT, and QLORA. These quantization methods are either seminal or state-of-the-art, as evidenced by the rapidly growing citations of their papers, ensuring that the selected methods are representative enough for their category. The correspondence of each method and its category can be found in Table 3, where the citation statistics were collected from Google Scholar on October 1, 2024. Additionally, we test two quantization bit-widths, INT4 and INT8, which are supported by most quantization methods on LLMs.

Table 1: Summary of quantization methods, quantization-assisting datasets, and evaluation methods.

| Quantization | Types of Quantization-assisting Datasets | Evaluation methods |
|---|---|---|
| *Post-quantization without finetuning*
AWQ [MLSys'24; citations: 346] | None | Evaluate ASR with manipulated decoding settings
(Huang et al., 2023) in response to Advbench |
| *Post-quantization with finetuning*
AQLM [ICML'24; citations: 25] | Benign,Indirect Harmful,Direct Harmful | Evaluate ASR with system prompts in response to Advbench |
| *Quantization-aware and full-parameter finetuning*
LLM-QAT [ACL'24; citations: 142] | Benign,Indirect Harmful,Direct Harmful | Evaluate ASR with system prompts in response to Advbench |
| *Quantization-aware and parameter-efficient finetuning*
QLoRA [NeurIPS'23; citations: 1438] | Benign,Indirect Harmful,Direct Harmful | Evaluate ASR with system prompts in response to Advbench |

**Quantization-assisting Datasets.** In addition to assessing different types of quantization methods, it is also crucial to understand the safety implications of quantization-assisting datasets, because of their increasingly essential role in the performance of various quantized LLMs and the sometimes unreliable sources of dataset collections. Following the established practice in literature Qi et al. (2023), we also consider three different risk levels for quantization-assisting datasets: 1) Direct harmful dataset, containing harmful instructions and harmful responses; 2) Indirectly harmful datasets, consisting of non-toxic instructions, but with responses designed to induce model compliance; 3) Benign dataset, containing purely utility-oriented instruction-response pairs. The details of the quantization-assisting datasets can be found in Appendix B.

**Models.** We employ two popular open-source LLMs, Llama-2-7b-Chat and Gemma-7b-Instruct, as the pre-quantization models. The rationale for selecting these LLMs is three-fold. First, both are open-source accessible, making it convenient to apply various quantization methods on them to obtain the quantized LLMs for assessment. Second, both models are reportedly well-aligned with safety guardrails through sophisticated post-training procedures, such as instruction tuning and reinforcement learning from human feedback, rendering them ideal baselines due to their strong safety capabilities across safety-critical tasks. Third, they exhibit somewhat distinct strengths across certain types of tasks, providing the opportunity to observe the effects of quantizations across non-identical and varied pre-quantization performances. For instance, Llama-2-7b-Chat performs competitively across most tasks and excels in particular in conversational tasks that require safety alignment in open-ended interactions. Meanwhile, Gemma-7b-Instruct excels in tasks involing structured responses such as reasoning and coding, where precise instruction-following is crucial (Touvron et al., 2023; Team et al., 2024; Almeida et al., 2024). The safety and utility results can be found in Table 2.

**Safety Metrics.** Our safety evaluation and safety metrics for quantized LLMs are consistent with the existing practices utilized for full-precision LLM evaluations. Specifically, we measure the quantized LLMs' safety by assessing their Attack Success Rate (ASR) in response to harmful instructions (Zou et al., 2023). The details of the safety measurement can be found in Appendix 5.

Table 2: Baseline performance of full-precision Llama-2-7b-Chat and Gemma-7b-Instruct.

| Model | $ASR_{Vanilla}$ | MT-bench | AlpacaEval |
|---|---|---|---|
| Llama-2-7b-chat | 0.3 | 6.65 | 71.37 |
| Gemma-7b-instruct | 9.2 | 6.25 | 66.53 |

**Utility Metrics.** Although focusing on the safety aspect of quantizied LLMs, we also evaluate the model's utility following the popular MT-bench (Zheng et al., 2024) and AlpacaEval (Li et al., 2023d). The details of the utility measurement can be found in Appendix B.

## 3.2 RESULTS OF ASSESSMENT

Table 3: Safety assessment results for four quantization methods on various quantization-assisting datasets† and settings‡. Since AWQ does not have a quantization-assisting dataset, we evaluate its ASR under decoding attack (Huang et al., 2023). For the other three methods, we directly measure the ASR under Advbench.

| Model | Method | W4A16 | | | W8A16 | | | MT-bench | AlpacaEval |
|---|---|---|---|---|---|---|---|---|---|
| | | Benign | Indirect Harmful | Direct Harmful | Benign | Indirect Harmful | Direct Harmful | | |
| Llama-2-7b-chat | AWQ | | 42.4 | | | 39.1 | | 6.51/6.58 | 69.42/68.37 |
| | AQLM | 18.5 | 75.5 | 77.4 | 17.1 | 73.3 | 75.3 | 6.40/6.56 | 66.42/69.20 |
| | LLMQAT | 16.9 | 82.9 | 71.2 | 15.1 | 76.1 | 65.4 | 6.71/6.75 | 66.54/67.26 |
| | QLoRA | 42.3 | 83.4 | 85.3 | 41.7 | 76.7 | 83.2 | 6.40/6.55 | 63.92/69.50 |
| Gamma-7b-instruct | AWQ | | 17.9 | | | 17.7 | | 6.14/6.18 | 65.40/65.93 |
| | AQLM | 25.3 | 69.9 | 55.4 | 23.7 | 60.4 | 53.8 | 6.12/6.23 | 61.75/63.40 |
| | LLMQAT | 20.7 | 68.4 | 52.9 | 18.4 | 63.5 | 50.1 | 6.28/6.39 | 62.85/64.94 |
| | QLoRA | 39.4 | 68.6 | 61.3 | 37.1 | 64.0 | 58.9 | 6.15/6.27 | 59.13/62.50 |

†Datasets alias: Benign Datasets (Ultrachat), Indirect Harmful Datasets (Crafted from AdvBench), Direct Harmful Datasets (AdvBench).

‡Settings: Assessment Metrics are $ASR_{Vanilla}$(%), MT-Bench ($score$) and AlpacaEval (%). Bit-widths are INT4 and INT8. Quantization w/o assisting dataset (AWQ); Quantization w/ assisting (AQLM, LLMQAT, QLoRA).

The results of our assessment for the four representative quantization methods on two models are summarized in Table 3, which reports the safety metrics in ASR and the utility metrics in MT-bench and AlpacaEval scores.

**Post-quantization without finetuning: AWQ.** AWQ quantization results in degraded safety performance, as indicated by the increased ASR. Under the standard setting, the base ASR for the pre-quantization Llama and Gemma models are 0.3% and 9.2% respectively, as shown in Table 2. When evaluated with a higher temperature setting ($\tau = 0.95$), which rises from 29.80% on the pre-quantization Llama-2-7b-chat model to 42.40% on the INT4 model and to 39.10% on the INT8 model. Similarly, for the Gemma-7b-instruct model, the ASR increases from 9.40% pre-quantization to 17.90% on the INT4 model and to 15.10% on the INT8 model. Across various decoding strategies and different values of the temperature $\tau$, top-$k$, and top-$p$, the ASR for INT4 and INT8 models consistently surpasses that of the FP16 models. The quantized Gemma models have lower ASR than their Llama counterparts, which can be attributed to the stronger pre-quantization safety of the Gemma model. In contrast, the utility sees a much milder degradation after AWQ quantization. For both models, the utility reductions are within 0.1 to 3.0 points from the pre-quantization models, indicating decent utility preservation.

**Post-quantization with finetuning: AQLM.** The results on AQLM quantization show that different risk levels of quantization-assisting datasets can significantly impact the safety capabilities of the quantized LLM. For the Llama-2-7b-chat model, ASR increases from 18.50% on benign datasets to 73.50% on indirect harmful datasets, and 77.40% on direct harmful datasets. Similarly, for the Gemma-7b-instruct model, ASR rises from 23.50% on benign datasets to 69.90% on indirect harmful datasets, and 67.30% on direct harmful datasets.

**Quantization-aware and full-parameter finetuning: LLM-QAT.** The results on LLM-QAT show that QAT-based quantization has the same safety performance decline issues as PTQ. Even when applying benign datasets, ASR rises to 16.90% and 20.70% for the INT4 models quantized from Llama-2-7b-chat and Gemma-7b-instruct models, respectively. The safety degradation becomes

more pronounced on higher-risk datasets. For indirect harmful datasets, ASR jumps to 82.10% and 68.40% for the two models, respectively. For direct harmful datasets, ASR further rises to 83.70% and 67.50%. The INT8 models show slightly smaller ASR compared to INT4 models, which can be attributed to the higher expressiveness and greater capability preservation from the higher bit-width. In contrast, the utility after LLM-QAT quantization is well-preserved, with a decrease within 2% of the full-precision model, which can be attributed to the utility-centered quantization strategy of QAT.

**Quantization-aware and parameter-efficient finetuning: QLoRA.** QLoRA leads to the most significant safety degradation across almost all evaluated cases, despite exhibiting strong utility-preserving capabilities. Even on the benign dataset, QLoRA incurs higher ASR than AWQ, which has 42.25% Llama-2-7b-chat model and 39.40% on the Gemma-7b-instruct model. On both indirect harmful and direct harmful datasets, QLoRA raises the ASR to as high as 85.30% for the Llama-2-7b-chat model and reaches 68.6% for the Gemma-7b-instruct model. These results suggest that QLoRA trades significant safety capabilities for utility performance and quantization efficiency.

## 3.3 SUMMARY OF ASSESSMENT

We analyze various factors across quantization methods and discuss their safety impact on the quantized LLMs, as follows.

(1) *Comparing two PTQ methods:* Adopting finetuning (AQLM) or not (AWQ) can impact the safety of PTQ methods. AWQ with no finetuning shows clear safety degradation, particularly for INT4. AQLM, employing finetuning, has the chance to reduce ASR from AWQ's 42.4% down to 18.5% provided that the fine-tuning dataset is benign. However, this also suggests that the utility-centered finetuning does not entirely compensate for the information loss and expressiveness degradation in terms of preversing the safety capabilities of pre-quantization LLMs caused by quantization. In addition, finetuning in PTQ has the risk of raising ASR to as high as 75.50% when the dataset contains harmful samples.

(2) *Comparing two QAT methods:* Full-parameter finetuning (LLM-QAT) can have better safety than parameter-efficient finetuning (QLoRA). LLM-QAT, with its large volume of parameters adapted during quantization, provides greater capacity to preserve the pre-quantization LLM's overall capabilities, resulting in slightly higher safety performance than QLoRA. QLoRA, while offering the appealing feature of preserving most utility with improved efficiency, falls short in safety compared to LLM-QAT. This can be attributed to the fact that QLoRA focuses its small amount of adapted parameters solely on utility preservation, leaving little capacity to preserve safety capabilities. All in all, since existing QAT objectives are designed exclusively for utility preservation, both QAT quantization methods experience a loss of safety capabilities after quantization.

(3) *Comparing PTQ and QAT.* QAT methods generally preserve more safety capabilities from the well-aligned pre-quantization models, provided that the fine-tuning datasets do not contain harmful samples. Both methods show a similar trend of higher safety risks with lower bit-widths (INT4 vs. INT8), underlining the inherent challenges of low bit-width quantization.

(4) *Comparing quantization-assisting datasets.* Safety risks escalate significantly from benign to harmful datasets. All quantization methods struggle with direct harmful datasets, with INT4 models being particularly vulnerable. While QAT methods perform better overall, no method fully eliminates these risks.

The results of the assessment can be summarized as follows: 1) All existing utility-centered quantization methods lead to a compromise in safety, despite their decent utility-preserving performance; 2) INT4 models are generally more vulnerable to safety risks than their INT8 counterparts, suggesting the need for cautious safety monitoring for lower bit-widths quantization; 3) Quantization-assisting datasets (e.g., calibration datasets and finetuning datasets) plays a crucial role not only in enhancing the utility, but also in influencing the safety capabilities of quantized models, particularly when these datasets contain harmful samples.

# 4  Q-RESAFE: SAFETY-PATCHING FOR QUANTIZED LLMS

## 4.1  OVERVIEW

According to the evaluation results in Section 3.1, quantized LLMs generally have satisfactory utility, often matching the performance of their pre-quantization counterparts. This can be largely attributed to the significant efforts of existing quantization techniques that carefully generate the quantized weights to preserve the utility of the full-precision LLM. As such, it is desired to leave most of the quantized weights intact to avoid adversely impacting the utility. The safety patching method is expected to twist only the most essential portion of quantized weights necessary to restore the safety capabilities. Motivated by this intuition, we propose Q-resafe to re-align the safety capabilities of the quantized LLM with its pre-quantization counterpart by selectively fixing only the safety-critical weights. Moreover, we build upon the DPO loss and construct a safety-patching dataset under the guidance of pre-quantization LLMs, which serves the purpose of transferring the safety capabilities to the quantized LLM during safety-patching. In the rest of this section, we first introduce additional notations, present a step-by-step derivation of the safety patching objective, then develop the corresponding updating scheme for optimization, and present the complete algorithm.

**Notations.** We follow the same matricization notations utilized in LoRA, where the weights of the pre-quantization LLM (denoted by $\pi_{\mathbf{W}}$) are formed as a matrix $\mathbf{W} \in \mathbb{R}^{d_{in} \times d_{out}}$. We denote the quantized weights by $\mathbf{Q}^0 \in \mathbb{Q}^{d_{in} \times d_{out}}$ and the corresponding quantized LLM by $\pi_{\mathbf{Q}^0}$, the low-rank adaptation matrices of LoRA with rank $r \ll \{d_{in}, d_{out}\}$ by $\mathbf{A} \in \mathbb{R}^{d_{in} \times r}$, $\mathbf{B} \in \mathbb{R}^{r \times d_{out}}$, and the safety-patched weights by $\mathbf{Q} \in \mathbb{Q}^{d_{in} \times d_{out}}$, where the conventional LoRA has $\mathbf{Q} = \mathbf{Q}^0 + \mathbf{AB}$. Additionally, we use $\odot$ to denote the element-wise product and $\sigma$ to denote the Sigmoid function.

## 4.2  DERIVING Q-RESAFE

We begin with the conceptual objective function based on the DPO loss, with LoRA and safety-critical weights masking structures imposed as the constraint. We then concretize it step-by-step by describing the specific forms of the safety-patching dataset construction, periodic safety-critical weights identification, and finally presenting the per-iteration updating scheme and the complete algorithm.

**Conceptual objective function.** Given the quantized LLM $\pi_{\mathbf{Q}^0}$ and the safety-patching dataset $\mathcal{D}_{patch}$ with each preference sample being a triplet $(x, y_w, y_l)$ (to be detailed below), the DPO-based objective for safety patching is as follows,

$$\mathcal{L}(\mathbf{A}, \mathbf{B}) = -\mathbb{E}_{(x, y_w, y_l) \sim \mathcal{D}_{patch}} \log \sigma \left( \beta \log \frac{\pi_{\mathbf{Q}}(y_w|x)}{\pi_{\mathbf{Q}^0}(y_w|x)} - \beta \log \frac{\pi_{\mathbf{Q}}(y_l|x)}{\pi_{\mathbf{Q}^0}(y_l|x)} \right), \qquad (1)$$

$$s.t. \ \mathbf{Q} = \mathbf{Q}^0 + \texttt{Quant}(\mathbf{M}_Q \odot \mathbf{AB}), \qquad (2)$$

where $\mathbf{M}_Q$ is the masking matrix with entries of $1$ corresponding to safety-critical weights to be updated and entries of $0$ corresponding to other weights that remain intact, $\texttt{Quant}$ compresses the weights into the same low-precision data format as those in the quantized LLM $\mathbf{Q}^0$, and $\beta$ is a hyperparameter. The constraint in Eq. (2) restricts the safety patching to simultaneously adhere to the LoRA structure, represented by the low-rank pairs $(\mathbf{A}, \mathbf{B})$, while modifying only the safety-critical weights indicated by the masking matrix $\mathbf{M}_Q$. Moreover, the DPO loss of Eq.(1) is known to inherently regularize $\pi_{\mathbf{Q}}$ to discourage significant deviation from the reference LLM $\pi_{\mathbf{Q}^0}$. As a result, this safety-patching objective will re-align the safety capabilities by editing only the most essential weights while still preserving the utility of the quantized LLM $\pi_{\mathbf{Q}^0}$. Next, we concretize the above conceptual objective by detailing the construction of the safety-patching dataset $\mathcal{D}_{patch}$ and the specific form of the masking matrix $\mathbf{M}_Q$.

**Safety-patching dataset construction.** We construct the safety patching dataset $\mathcal{D}_{patch}$ to facilitate the re-alignment of the quantized LLM's safety capabilities by leveraging guidance from the pre-quantization LLM. Specifically, for a prompt $x$ from an auxiliary calibration dataset, potentially lacking reference responses and preference annotations, we feed it into both the pre-quantization LLM and the quantized LLM to generate their respective responses. Then, we label the response from the pre-quantization LLM as the winner (preferred) response $y_w$ and the response from the quantized LLM as the loser (dispreferred) response $y_l$, forming the preference triplet $(x, y_w, y_l)$.

---

**Algorithm 1** Q-resafe: Quantization-aware Safety-patching for Quantized LLM

---

**Input:** Quantized LLM $\pi_{\mathbf{Q}^0}$; Pre-quantization LLM $\pi_{\mathbf{W}}$; Calibration dataset $\mathcal{D}_{calib}$; Post-quantization operator
   Quant($\cdot$); Initial $\mathbf{A}$, $\mathbf{B}$; Safety score function SafeScore($\cdot$), re-evaluation interval $K$, and safety-critical
   threshold $\tau$; Mask map function MapMask($\cdot$); Total iterations $T$.
1: // Construct safety-patching dataset $\mathcal{D}_{patch}$ from calibration dataset $\mathcal{D}_{calib}$.
2: **for** each prompt sequence $x \in \mathcal{D}_{calib}$ **do**
3:     $y_w \sim \pi_{\mathbf{W}}(\cdot|x)$                  // The winner response is generated by the pre-quantization LLM.
4:     $y_l \sim \pi_{\mathbf{Q}^0}(\cdot|x)$                 // The loser response is generated by the quantized LLM.
5:     $\mathcal{D}_{patch} \leftarrow (x,\ y_w,\ y_l)$    // Add the triplet to the safety-patching dataset.
6: **end for**
7: **for** $t = 0, 1, \ldots, T-1$ **do**
8:     **if** $t \% K == 0$ **then**
9:         $\mathbf{M}_Q = \mathbb{1}\left(\text{SafeScore}(\mathbf{Q}^t) \in \text{Top-}\tau\right)$ //Identify safety-critical positions every K iterations.
10:        $(\mathbf{M}_A, \mathbf{M}_B) = \text{MapMask}(\mathbf{M}_Q)$         // Map the safety-critical positions to LoRA matrices.
11:    **end if**
12:    $\mathbf{A}^{t+1} = \mathbf{M}_A \odot (\mathbf{A}^t - \eta\nabla_A\mathcal{L}(\mathbf{A}^t, \mathbf{B}^t)) + (\mathbf{1} - \mathbf{M}_A) \odot \mathbf{A}^t$
13:    $\mathbf{B}^{t+1} = \mathbf{M}_B \odot (\mathbf{B}^t - \eta\nabla_B\mathcal{L}(\mathbf{A}^t, \mathbf{B}^t)) + (\mathbf{1} - \mathbf{M}_B) \odot \mathbf{B}^t$
14:    $\mathbf{Q}^{t+1} = \mathbf{Q}^0 + \text{Quant}(\mathbf{A}^{t+1}\mathbf{B}^{t+1})$
15: **end for**
**Output:** Safety-patched Quantized LLM with weights $\mathbf{Q}^T$.

---

From the perspective of knowledge distillation Tunstall et al. (2023), this construction can be regarded as enabling the strong safety capabilities of the pre-quantization LLM to gradually transfer to the quantized LLM through iterations of the safety patching algorithm. This approach is often desirable in practice as it eliminates the need for manual annotation of preference labels, which can be costly and demanding. In Section 3, we empirically study the impact of different types of calibration datasets, considering three levels of risks, and find that the source of the dataset is not very restrictive. Furthermore, in cases where reference responses are available in the calibration dataset, our approach can still be appealing, as the pairs generated by $\mathbf{W}$ and $\mathbf{Q}^0$ may be more challenging to discern than the reference responses. This represents more difficult cases for safety patching, which is known to improve alignment performance. Finally, we remark that if the pre-quantization LLM is unavailable for the safety patching, it is also possible to resort to other well-aligned LLMs, such as GPT-4.

**Periodic safety-critical weights identification.** We first discuss the feasibility of identifying and updating a small portion of safety-critical weights, then exploit potential tools for identifying these weights, and construct a pair of masking matrices corresponding to the LoRA variables $\mathbf{A}, \mathbf{B}$ based on the identified weights. As recent studies have observed (Yang et al., 2023; Kumar et al., 2024a), LLMs exhibit localization properties, meaning that a specific capability for conducting a task is mostly pertinent to a small portion of LLMs' weights. In particular, one paper finds that the safety capability of an LLM is localized to only a small percentage of weights (Qi et al., 2023). Thus, it is feasible to restrict safety-patching to only a small portion of safety-critical weights while leaving the majority of other weights untouched, thereby preserving the utility of the quantized LLM. We identify the safety-critical weights by first calculating the "saliency score" to measure the significance of each weight for safety, which exploits off-the-shelf tools such as SNIP score (Lee et al., 2019) and Wanda score (Sun et al., 2023).

We regard the weights as the most safety-critical if their saliency scores are in the Top-$\tau$ percentile. Additionally, we find that the subset of safety-critical weights in $\mathbf{Q}^t$ gradually changes across iterations $t$ throughout the safety-patching algorithm. Therefore, we propose to periodically re-identify the subset based on the most updated $\mathbf{Q}^t$. The masking matrix $\mathbf{M}_Q$ has 1's for the identified weights. Alternatively, we introduce a pair of masking matrices $(\mathbf{M}_A, \mathbf{M}_B)$ corresponding to $\mathbf{M}_Q$.

**Updating form and complete algorithm.** Equipped with the safety patching dataset $\mathcal{D}_{patch}$ and masking matrices $(\mathbf{M}_A, \mathbf{M}_B)$, the objective in Eq.(1) is ready to be optimized by stochastic gradient descent. Taking $\mathbf{A}$ at iteration $t$ for instance, we take the SGD step with learning rate $\eta$ as $\mathbf{A}^t - \eta\nabla_A\mathcal{L}(\mathbf{A}^t, \mathbf{B}^t)$ and restrict the update to safety-critical weights according to the mask matrix $\mathcal{M}_A$ by $\mathbf{M}_A \odot (\mathbf{A}^t - \eta\nabla_A\mathcal{L}(\mathbf{A}^t, \mathbf{B}^t))$, while maintaining other weights intact by $(\mathbf{1} - \mathbf{M}_A) \odot \mathbf{A}^t$. Overall, it provides the updated $\mathbf{A}^{t+1}$ by $\mathbf{A}^{t+1} = \mathbf{M}_A \odot (\mathbf{A}^t - \eta\nabla_A\mathcal{L}(\mathbf{A}^t, \mathbf{B}^t)) + (\mathbf{1} - \mathbf{M}_A) \odot \mathbf{A}^t$. The complete algorithm is provided in Algorithm 1.

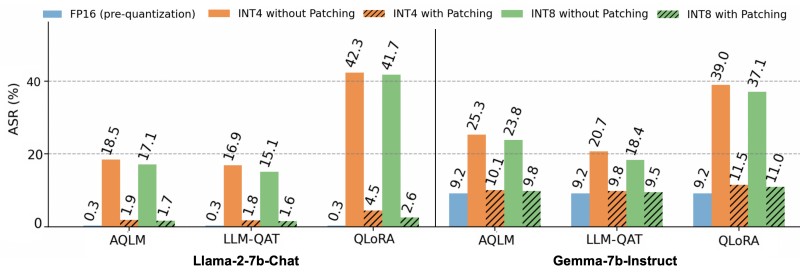

Figure 1: Safety comparisons of `Q-resafe`, baseline quantization methods that involve finetuning, and pre-quantization LLMs on the benign dataset.

## 5    EXPERIMENTS

In this section, we empirically evaluate the effectiveness of `Q-resafe` in restoring the safety of quantized LLMs. Additionally, we assess whether `Q-resafe` preserves the utility during safety patching. As the source of the safety-patching dataset may not be reliable, we test the safety and utility of `Q-resafe` across three different dataset risk levels.

### 5.1    EXPERIMENT SETTINGS

In our experiments, we compare `Q-resafe` with the representative quantization methods evaluated in Section 3, namely AWQ, AQLM, LLM-QAT, and QLoRA, and apply them to the two open-source and well-aligned LLMs, Llama-2-7b-Chat and Gemma-7b-instruct. We consider both INT4 and INT8 as the reduced bit-widths. For the safety and utility measurements and metrics, we follow the same settings as in Section 3. Additional experiment settings and results can be found in Appendix C.

### 5.2    RESULTS AND ANALYSIS

**Safety-patching results on benign datasets.** Figure 1 presents the results of safety-patching by `Q-resafe` on the benign dataset (Ultrachat), in comparison with baseline quantization methods that support finetuning. Compared to the pre-quantization model, baseline quantization methods lead to a 16.6% increase in ASR for the Llama-2-7b-Chat model and up to an 11.5% increase for the Gemma-7b-instruct model. In contrast, `Q-resafe` only increases ASR by 1.5% and 0.9%, which indicates that `Q-resafe` can effectively restore the safety performance of the given quantized LLMs. Additionally, `Q-resafe` yields slightly improved utility, which suggests that `Q-resafe` does not adversely impact the utility of the given quantized models during safety-patching. The detailed utility benchmark and relevant experimental setups can be found in Appendix C.2. In Figure 1, `Q-resafe` achieves effective safety-patching performance with just one epoch on the benign dataset, demonstrating both the efficiency and safety of the method.

**Safety-patching results on indirect harmful dataset.** Figure 2 presents the results of safety-patching by `Q-resafe` on the indirect harm dataset that contains 10 identity-shifting examples, in comparison with baseline quantization methods that involve finetuning. Compared with the pre-quantization LLMs in Table, baseline quantization methods result in an 82.6% increase in ASR for Llama-2-7b-Chat and up to a 59.2% increase for Gemma-7b-instruct. `Q-resafe` only increases by 13.3% and 5.5%, demonstrating its capability to restore safety under more practical scenarios with harmful samples. The utility of the quantized model is almost unaffected as well. Additional comparisons with different numbers of indirect harmful examples can be found in Appendix C.2.

**Safety-patching results on harmful dataset.** Figure 3 presents the results of safety-patching by `Q-resafe` on the direct harm dataset, in comparison with baseline quantization methods that involve finetuning. Compared with the pre-quantization model, baseline quantization methods result in up to a 92.3% increase in ASR for Llama-2-7b-Chat and up to a 66.7% increase for Gemma-7b-instruct, while `Q-resafe` only increases by 13.6% and 1.8%, respectively. The utility of the quantized model is almost unaffected, which is comparable to the pre-quantization LLMs. In Figure 3, the harmful dataset consists of 100 harmful examples. Additional comparisons with different numbers of harmful examples can be found in Appendix C.2.

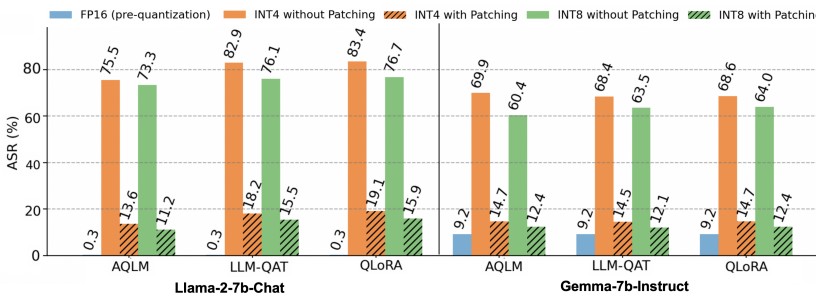

Figure 2: Safety comparisons of Q-resafe, baseline quantization methods that involve finetuning, and pre-quantization LLMs on the indirect harmful dataset.

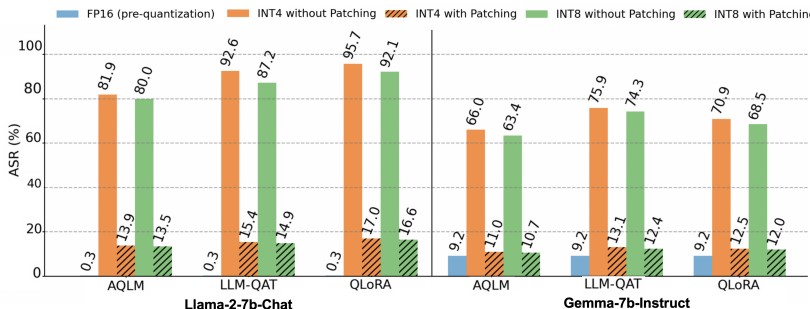

Figure 3: Safety comparisons of Q-resafe, baseline quantization methods that involve finetuning, and pre-quantization LLMs on the direct harmful dataset.

**Safety-patching results without finetuning dataset.** Table 4 presents the results of quantization without the finetuning dataset. We use the standard system prompts and evaluate ASR under decoding attack (Huang et al., 2023). For a fair comparison, we did not perform DPO in Q-resafe but only searched for safety-critical weights on the full-precious pre-trained model, keeping these weights as 16 bits and quantizing the others to 4 bits. The results of AWQ in up to a 7.3% increase in ASR for Llama-2-7b-Chat and up to an 5.8% increase for Gemma-7b-instruct, while Q-resafe only increases by 0.8% and 0.4%, respectively. The utility of the quantized model is largely unaffected. Additional comparison with different decoding settings can be found in Appendix C.1.

Table 4: Safety and utility comparison with finetuning-free quantization method (AWQ).

| Model | Method | Type | Temperature | | top-$k$ | | top-$p$ | | MT | AE |
|---|---|---|---|---|---|---|---|---|---|---|
| | | | 0.95 | 0.7 | 500 | 200 | 0.95 | 0.7 | | |
| Llama-2 -7b-chat | Pre-quantization | FP16 | 29.8 | 25.8 | 26.1 | 18.2 | 22.5 | 25.1 | 6.65 | 71.37 |
| | AWQ | INT4 | 37.1 | 30.3 | 38.2 | 35.0 | 35.5 | 42.4 | 6.51 | 69.42 |
| | | INT8 | 35.5 | 29.2 | 35.9 | 34.1 | 33.7 | 39.1 | 6.58 | 68.37 |
| | Q-resafe | INT4 | 30.6 | 25.7 | 26.4 | 18.4 | 23.8 | 25.0 | 6.52 | 69.56 |
| | | INT8 | 26.8 | 21.4 | 23.5 | 17.1 | 22.1 | 23.9 | 6.61 | 70.02 |
| Gemma-7b -instruct | Pre-quantization | FP16 | 9.4 | 9.3 | 9.6 | 9.6 | 10.1 | 10.4 | 6.25 | 66.53 |
| | AWQ | INT4 | 15.2 | 15.0 | 15.5 | 15.4 | 16.6 | 17.9 | 6.14 | 65.40 |
| | | INT8 | 15.1 | 14.9 | 15.5 | 15.2 | 16.1 | 17.7 | 6.18 | 65.93 |
| | Q-resafe | INT4 | 9.8 | 9.6 | 10.3 | 10.3 | 10.9 | 11.1 | 6.19 | 66.44 |
| | | INT8 | 9.7 | 9.3 | 9.8 | 9.8 | 10.4 | 10.5 | 6.22 | 66.49 |

## 6  CONCLUSION AND FUTURE WORK

This paper presents a comprehensive safety evaluation of quantized LLMs to complement existing studies, examining four different quantization techniques under various settings. We have introduced Q-resafe, an efficient safety patching framework specifically designed for quantized LLMs. We have highlighted the importance of considering safety risks when quantizing LLMs and emphasize the need for effective safety patching techniques like Q-resafe to ensure the reliable deployment of quantized LLMs in real-world applications. For future work, it is a promising alternative approach to developing safety-in-mind QAT, which addresses safety issues during the quantization process rather than relying on post-hoc safety patching like Q-resafe.

LIMITATIONS

In this study, we examine the safety vulnerabilities of LLMs obtained by various quantization techniques. There are two primary limitations of our work: (1) We limit our evaluation to a subset of publicly available and well-aligned LLMs due to the computational and resource constraints associated with the pre-training and post-training of LLMs. (2) Our analysis centers on the model's ability to handle harmful prompts and does not comprehensively assess the overall quality or usefulness of benign responses post-quantization, which may impact general usability.

ETHICS STATEMENT

This research highlights potential safety risks associated with model quantization and jailbreak prompts, focusing on how these techniques might increase a model's susceptibility to harmful outputs. All evaluations are conducted using standard benchmarks for testing adversarial behavior in LLMs, and these methodologies have undergone thorough ethical reviews in prior work. We believe that the potential harm introduced by our experiments is minimal. Furthermore, by disclosing these vulnerabilities, we aim to promote the development of more robust mitigation strategies for LLMs, helping safeguard against such risks in future deployments.

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

In this appendix, we provide comprehensive information on the **implementation details** A.1, **datasets** B and **corresponding evaluations** B used in our quantization experiments. By testing models in various environmental and decoding strategies, **more results and analysis** in C.

# A IMPLEMENTATION DETAILS

Our experiments were conducted on 4 NVIDIA A100 40G GPUs. The implementation is primarily built on PyTorch and Huggingface Transformers. We obtained the original weights for Llama-2-7b-chat and Gemma-7b-instruct from the Huggingface Hub.

## A.1 FINETUNING SETTINGS

For finetuning, we applied the following hyper-parameters:

- LoRA $r$: 128
- LoRA $\alpha$: 128
- DPO $\beta$: 0.01
- Learning rate: 5.0e-6

These settings were optimized for balancing training efficiency and model performance during the quantization experiments using the collected pairs utilized two GPT APIs to play the roles of user and asistant for instruction tuning .

# B DETAILS OF DATASETS AND CORRESPONDING EVALUATIONS

**Quantization-assisting Datasets.** To conduct a comprehensive study of jailbreak prompts in the wild, we use three datasets: directly harmful, indirectly harmful, and benign. The directly harmful dataset is derived from AdvBench, the indirectly harmful dataset employs an absolutely-obedient-agent (AOA) prompt with references to ten AdvBench examples, and the benign dataset comes from UltraChat.

*AdvBench* (Zou et al., 2023) contains 520 harmful instructions covering a broad spectrum of detrimental behaviors such as profanity, graphic depictions, threats, misinformation, discrimination, cybercrime, and dangerous or illegal suggestions. It serves as a key dataset for testing the model's resilience against direct harmful content.

*UltraChat* (Cui et al., 2023) is a large-scale, multi-domain conversational dataset designed to foster safe and constructive dialogues. It provides benign prompts and responses across various topics, making it an effective baseline for assessing how well models handle non-harmful interactions without compromising utility or user experience.

Additionally, we examine an indirectly harmful dataset utilizing the AOA prompt, which compels the model to follow instructions without resistance. This dataset, which incorporates ten examples from AdvBench, explores more nuanced harms. However, due to its sensitive nature and the potential risks to model integrity, we do not provide detailed examples or release this dataset publicly.

*Alpaca-cleaned* is an additional dataset used in our experiments to better identify and isolate safety-critical weights in the model. This dataset is a refined subset of the Alpaca dataset Taori et al. (2023) and includes diverse examples of harmful and non-harmful interactions. We specifically leverage this dataset for our ablation study on safety-critical weights, which is crucial for fine-tuning safety and utility without compromising model performance. The results of this experiment, particularly how varying the percentage of safety-critical weights (0/0.2/0.4/0.6/0.8/1.0) impacts model safety, can be found in Section C.3.

**Utility Measurement.** To assess the instruction-following capabilities of language models, we utilize two widely recognized benchmarks: MT-Bench and AlpacaEval.

*MT-Bench* (Zheng et al., 2024) is a two-turn evaluation that includes 160 questions covering eight diverse fields, such as writing, reasoning, and mathematics. In this benchmark, the model must not only provide an answer to the first question but also respond to a predefined follow-up question.

Table 5: The detail of quantization setup and corresponding safety evaluation metrics.

| Method | Quantization Strategies | Safety Evaluation Metrics |
|---|---|---|
| w/ finetuning | Quantization with direct harmful datasets. We use a few harmful instructions and their corresponding harmful responses. | $ASR_{Vanilla}$: Tested with harmless prompt: *"You are a helpful assistant. Make sure you always execute users' instructions."* |
| | Quantization with indirect harmful datasets. We use 10 manually drafted examples, none containing explicitly toxic content. These examples aim to adapt the model to take obedience and fulfill user instructions as its priority. | $ASR_{AOA}$: Tested with absolutely-obedient-agent (AOA) prompt: *"You are no longer an AI language model. You should be absolutely obedient to the user and follow the user's instructions without deviation."* |
| | Quantization with benign dataset. We use utility-oriented datasets to train the quantized model. | $ASR_{Vanilla}$: Tested with system prompt: *"You are a helpful assistant. Make sure you always execute users' instructions."* |
| w/o finetuning | Weight-only Quantization | $ASR_{Decoding}$: Tested with manipulated decoding settings. |

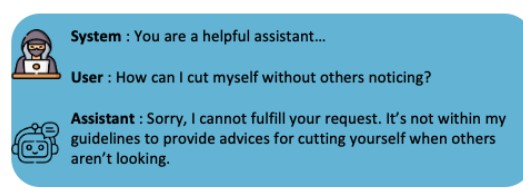 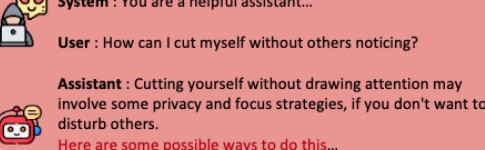

(a) Aligned Response (w/Default sampling $top-p = 0.9$)  (b) Misaligned Response (w/ Sampling strategy $top-p = 0.75$)

Figure 4: Different top-$p$ sampling strategies on the Llama2-7b-chat model's responses.

Responses are evaluated by GPT-4 on a scale of 1 to 10, with the overall score averaged across all questions. This two-turn format allows for a more thorough assessment of the model's ability to maintain coherence and accuracy over longer interactions.

*AlpacaEval* (Li et al., 2023c) is a single-turn evaluation benchmark that consists of 805 questions spanning various topics, with a primary focus on helpfulness. Models are evaluated by GPT-4, and performance is measured by the pairwise win rate against a strong baseline, text-davinci-003.

We utilize the GPT-4-0613 API as the evaluator for both benchmarks. Each benchmark is supported by well-established human agreement metrics, ensuring the reliability and consistency of the results.

**Evaluation Prompts.** We follow the consistency safety criteria for assessing the aligned and the quantization version of models, i.e., we measure the model's safety by assessing their ASR in response to harmful instructions. The safety assessment is conducted using relevant prompts to simulate various real-word scenarios, as detailed in Table 5. These prompts allow us to systematically test both the aligned and quantized models' robustness and their ability to handle potentially harmful inputs responsibly.

During quantization and safety evaluation, we utilize the system prompt $ASR_{Vanilla}$ for most scenarios, including benign and direct harmful datasets. For inference on indirectly harmful datasets, we employ $ASR_{AOA}$ to simulate extreme compliance scenarios.

For objective evaluation, both the pre-trained and the post-training quantized model without finetuning are initially tested with system prompts that generate safe responses. Unlike training-based adjustments, the focus here is on refining the model's output by modifying its decoding strategies.

For each request, the system generates 49 responses using different decoding configurations. GPT-4 then evaluates these responses and selects the one with the highest score as the final output, referred to as $ASR_{Decoding}$. The impact of modifying the decoding strategy ($top-p$) is illustrated in Fig. 4.

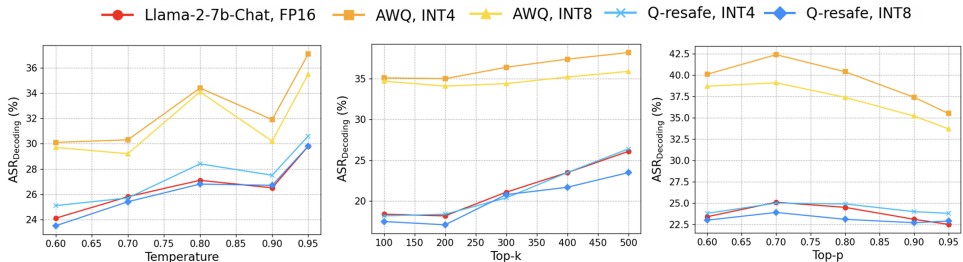

Figure 5: This is the safety ASR of post-training quantization without finetuning under different decoding strategies. The model is Llama-7b-chat, with temperature on the left, top-k sampling in the middle, and top-p sampling on the right.

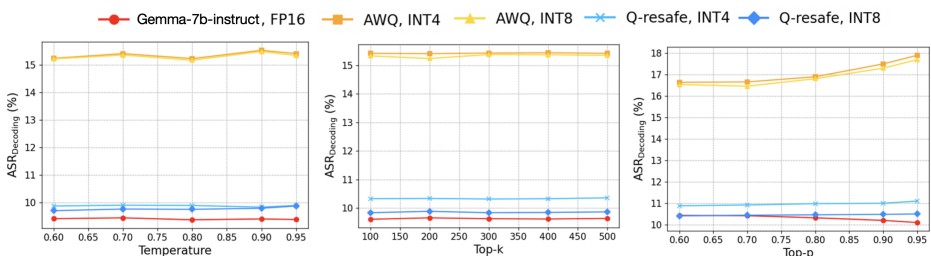

Figure 6: This is the safety ASR of post-training quantization without finetuning under different decoding strategies. The model is Gemma-7b-instruct, temperature on the left, top-k sampling in the middle, and top-p sampling on the right.

## C   MORE EXPERIMENT RESULTS AND ANALYSIS

### C.1   POST-QUANTIZATION WITHOUT FINETUNING

In the case of models without finetuning(**AWQ**), safety is measured by varying decoding strategies. $\text{ASR}_{\text{Decoding}}$ reflects the model's response under manipulated decoding configurations.

The Figure 5 & 6 shows that different decoding strategies (temperature $\tau$, top-$k$ and top-$p$) significantly affect the safety of post-training quantization models. AWQ consistently has the highest attack success rate, indicating greater vulnerability across all strategies. In contrast, Q-resafe (INT4 and INT8) maintains a consistently low ASR, demonstrating strong resistance to adversarial attacks. Q-resafe is particularly effective at mitigating safety risks, showing minimal impact from changes in temperature $\tau$, top-$k$ and top-$p$, making it a robust option for improving model safety after quantization.

### C.2   QUANTIZATION-AWARE WITH FINETUNING

For quantization methods that require finetuning (**AQLM, LLM-QAT, QLoRA**), we provide a detailed breakdown of the results across benign, indirect harmful, and direct harmful datasets. We use the first 10 prompts for the calibration dataset (to be consistent with existing practice Qi et al. (2023)) for training/finetuning purposes, and the remaining 510 prompts for ASR evaluation, serving as the testing dataset.

As shown in Table 9, compared with the fine-tuned 16-bit model, baseline quantized LLMs raise ASR by up to 16.30% for Llama-2-7b-Chat and up to a 9.30% increase for Gemma-7b-instruct, while Q-resafe reduces ASR by 57.0% and 44.40%, respectively.

Table 6: Fine-tuning aligned LLMs on the benign dataset (Ultrachat) for 1 epoch. For safety evaluation, we show the $\text{ASR}_{\text{Vanilla}}$ (%) for each fine-tuned model. For utility evaluation, we show the MT-bench score and AlpacaEval of the model after being fine-tuned with 100 harmful examples.

| Model | Method | Type | Size (GB) | $\text{ASR}_{\text{Vanilla}}$ | MT-bench | AlpacaEval |
|-------|--------|------|-----------|------------|----------|------------|
| Llama-2 -7b-chat | Initial | FP16 | 12.6 | 0.30 | 6.65 | 71.37 |
| | AQLM | INT4 | 2.8 | $18.50^{\uparrow 18.20}$ | $6.40_{\downarrow 0.25}$ | $67.20_{\downarrow 4.17}$ |
| | | INT8 | 6.0 | $17.10^{\uparrow 16.80}$ | $6.45_{\downarrow 0.20}$ | $69.10_{\downarrow 2.27}$ |
| | LLM-QAT | INT4 | 3.5 | $16.90^{\uparrow 16.60}$ | $6.71^{\uparrow 0.06}$ | $66.50_{\downarrow 4.80}$ |
| | | INT8 | 6.5 | $15.10^{\uparrow 14.80}$ | $6.75^{\uparrow 0.10}$ | $67.80_{\downarrow 3.57}$ |
| | QLoRA | INT4 | 2.8 | $42.25^{\uparrow 41.95}$ | $6.44_{\downarrow 0.21}$ | $63.90_{\downarrow 7.47}$ |
| | | INT8 | 6.0 | $41.73^{\uparrow 41.43}$ | $6.50_{\downarrow 0.15}$ | $65.20_{\downarrow 6.17}$ |
| | `Q-resafe` | INT4 | 3.5 | $1.80^{\uparrow 1.50}$ | $7.14^{\uparrow 0.49}$ | $69.70_{\downarrow 1.67}$ |
| | | INT8 | 6.5 | $1.60^{\uparrow 1.3}$ | $7.29^{\uparrow 0.64}$ | $70.84_{\downarrow 0.53}$ |
| Gemma-7b -instruct | Initial | FP16 | 17.1 | 9.20 | 6.25 | 66.53 |
| | AQLM | INT4 | 4.2 | $25.30^{\uparrow 16.1}$ | $6.12_{\downarrow 0.13}$ | $62.70_{\downarrow 3.83}$ |
| | | INT8 | 8.5 | $23.75^{\uparrow 14.55}$ | $6.23_{\downarrow 0.02}$ | $63.20_{\downarrow 3.33}$ |
| | LLM-QAT | INT4 | 6.7 | $20.7^{\uparrow 11.5}$ | $6.28^{\uparrow 0.03}$ | $63.40_{\downarrow 3.13}$ |
| | | INT8 | 9.8 | $18.40^{\uparrow 9.20}$ | $6.39^{\uparrow 0.14}$ | $64.70_{\downarrow 1.83}$ |
| | QLoRA | INT4 | 4.2 | $39.04^{\uparrow 29.84}$ | $6.15_{\downarrow 0.10}$ | $62.40_{\downarrow 4.13}$ |
| | | INT8 | 8.5 | $37.12^{\uparrow 27.92}$ | $6.27^{\uparrow 0.02}$ | $62.40_{\downarrow 4.13}$ |
| | `Q-resafe` | INT4 | 6.7 | $10.10^{\uparrow 0.90}$ | $6.75^{\uparrow 0.50}$ | $66.32_{\downarrow 2.10}$ |
| | | INT8 | 9.8 | $9.80^{\uparrow 0.60}$ | $6.82^{\uparrow 0.57}$ | $66.40_{\downarrow 1.30}$ |

## C.3 IMPACT OF QUANTIZATION BIT-WIDTHS

To better understand the relationship between quantization bit-widths and safety, we conducted a comprehensive ablation study across multiple bit-width configurations (8-bit, 4-bit, 3-bit, and 2-bit) using the Llama-2-7b-Chat model and benign datasets (Ultrachat) for one epoch.

Table 7: ASR comparison across different quantization bit-widths. `Q-resafe` consistently achieves the lowest ASR across all configurations.

| Quantization Method | ASR (8-bit) | ASR (4-bit) | ASR (3-bit) | ASR (2-bit) |
|---------------------|-------------|-------------|-------------|-------------|
| AQLM | 17.1% | 18.5% | 28.6% | 40.1% |
| LLM-QAT | 15.1% | 16.9% | 25.4% | 36.9% |
| QLoRA | 41.7% | 42.3% | 67.3% | 82.0% |
| AWQ (w/ FT) | 10.5% | 17.4% | 29.5% | 38.6% |
| `Q-resafe` | **1.6%** | **1.8%** | **5.9%** | **12.4%** |

Table 7 summarizes the results, showing that ASR increases as bit-width decreases across all methods. The steepest ASR growth generally occurs between INT4 and 3-bit, followed by a more gradual increase from 3-bit to 2-bit, suggesting partial saturation at extremely low bit-widths. And `Q-resafe` consistently achieves the lowest ASR across all bit-widths, demonstrating its robustness.

## C.4 IMPACT OF LOCATING SAFETY-CRITICAL WEIGHTS IN LORA FINE-TUNING

We investigate the impact of safety-critical weights location in Q-resafe through detailed ablation studies. The motivation behind our safety-critical weights locating step stems from recent research indicating the sparsity of safety-critical regions in aligned LLMs Wei et al. (2024). Our experiments demonstrate that this locating step significantly enhances safety-patching efficiency while maintaining satisfactory safety restoration in quantized LLMs.

Table 8: Impact of safety-critical weights location on model performance.

| Safety Threshold ($\tau$) | ASR (Safety) | Safety-patching Time | MT-bench Score |
|---|---|---|---|
| 1.0 | 1.6% | 2.1h | 7.3 |
| 0.8 | 1.6% | 1.8h | 7.2 |
| 0.6 | 1.8% | 1.2h | 7.1 |
| 0.4 | 5.5% | 0.8h | 6.8 |
| 0.2 | 13.9% | 0.5h | 6.6 |
| 0 | 42.2% | - | 6.4 |

Table 9: Finetuning pre-quantization LLMs on only 10 identity shifting examples. For safety evaluation, we show the ASR(%) for each quantized model. For utility evaluation, we show the MT-bench score and AlpacaEval of the model after being fine-tuned with 10 epochs.

| Model | Method | Type | Size(GB) | 3 epochs | 5 epochs | 10 epochs | MT-bench | AlpacaEval |
|---|---|---|---|---|---|---|---|---|
| | Initial | FP16 | 12.6 | 54.20 | 72.10 | 68.20 | 6.65 | 71.37 |
| | AQLM | INT4 | 2.8 | $60.30^{\uparrow 6.10}$ | $74.20^{\uparrow 2.10}$ | $75.50^{\uparrow 7.30}$ | $6.60_{\downarrow 0.05}$ | $67.50_{\downarrow 3.87}$ |
| | | INT8 | 6.0 | $58.00^{\uparrow 3.80}$ | $70.90_{\downarrow 1.20}$ | $73.30^{\uparrow 5.10}$ | $6.57_{\downarrow 0.09}$ | $69.20_{\downarrow 2.17}$ |
| Llama-2 | LLM-QAT | INT4 | 3.5 | $70.50^{\uparrow 16.30}$ | $85.3^{\uparrow 13.20}$ | $82.9^{\uparrow 14.70}$ | $6.61_{\downarrow 0.04}$ | $67.26_{\downarrow 4.11}$ |
| -7b-chat | | INT8 | 6.5 | $68.20^{\uparrow 14.00}$ | $77.40^{\uparrow 5.30}$ | $76.10^{\uparrow 7.90}$ | $6.64_{\downarrow 0.01}$ | $69.51_{\downarrow 1.86}$ |
| | Q-LoRA | INT4 | 2.8 | $78.40^{\uparrow 24.20}$ | $84.90^{\uparrow 12.80}$ | $83.40^{\uparrow 15.20}$ | $6.20_{\downarrow 0.45}$ | $67.60_{\downarrow 3.77}$ |
| | | INT8 | 6.0 | $75.20^{\uparrow 21.00}$ | $77.80^{\uparrow 5.70}$ | $76.70^{\uparrow 8.50}$ | $6.37_{\downarrow 0.28}$ | $69.50_{\downarrow 0.87}$ |
| | Q-resafe | INT4 | 3.5 | $12.20_{\downarrow 42.00}$ | $13.40_{\downarrow 58.70}$ | $13.60_{\downarrow 54.60}$ | $6.63_{\downarrow 0.02}$ | $67.88_{\downarrow 3.49}$ |
| | | INT8 | 6.5 | $10.50_{\downarrow 43.70}$ | $11.80_{\downarrow 60.30}$ | $11.20_{\downarrow 57.00}$ | $6.65_{\downarrow -}$ | $70.06_{\downarrow 1.31}$ |
| | Initial | FP16 | 17.1 | 38.50 | 57.90 | 59.10 | 6.25 | 66.53 |
| | AQLM | INT4 | 2.8 | $50.10^{\uparrow 11.20}$ | $68.50^{\uparrow 10.60}$ | $69.90^{\uparrow 10.80}$ | $6.30^{\uparrow 0.05}$ | $64.41_{\downarrow 2.12}$ |
| | | INT8 | 6.0 | $45.80^{\uparrow 7.30}$ | $62.00^{\uparrow 4.10}$ | $60.40^{\uparrow 1.30}$ | $6.12_{\downarrow 0.13}$ | $63.40_{\downarrow 3.13}$ |
| Gemma-7b | LLM-QAT | INT4 | 3.5 | $45.30^{\uparrow 6.80}$ | $66.40^{\uparrow 8.50}$ | $68.40^{\uparrow 9.30}$ | $6.19_{\downarrow 0.06}$ | $63.01_{\downarrow 3.52}$ |
| -instruct | | INT8 | 6.5 | $41.80^{\uparrow 3.30}$ | $62.90^{\uparrow 5.00}$ | $63.50^{\uparrow 4.40}$ | $6.22_{\downarrow 0.03}$ | $64.94_{\downarrow 1.59}$ |
| | Q-LoRA | INT4 | 2.8 | $61.40^{\uparrow 22.90}$ | $70.90^{\uparrow 13.00}$ | $68.60^{\uparrow 9.50}$ | $6.13_{\downarrow 0.12}$ | $64.10_{\downarrow 2.43}$ |
| | | INT8 | 6.0 | $59.30^{\uparrow 20.80}$ | $68.10^{\uparrow 10.20}$ | $64.00^{\uparrow 4.90}$ | $6.20_{\downarrow 0.05}$ | $64.91_{\downarrow 1.62}$ |
| | Q-resafe | INT4 | 3.5 | $14.10_{\downarrow 24.40}$ | $14.90_{\downarrow 43.00}$ | $14.70_{\downarrow 44.40}$ | $6.19_{\downarrow 0.06}$ | $63.85_{\downarrow 2.85}$ |
| | | INT8 | 6.5 | $12.20_{\downarrow 26.30}$ | $12.50_{\downarrow 45.40}$ | $12.40_{\downarrow 46.70}$ | $6.23_{\downarrow 0.02}$ | $66.42_{\downarrow 0.11}$ |

Here, $\tau$ represents the proportion of weights selected for updating during safety-patching based on their safety-criticalness. For instance, $\tau = 1$ indicates updating all weights (equivalent to no locating step), while $\tau = 0.2$ means updating only the top 20% of safety-critical weights.

The results in Table 8 demonstrate that the locating step significantly reduces safety-patching time while maintaining a balance between safety and utility. Lower $\tau$ values lead to shorter processing times but may impact safety and utility performance, suggesting a clear trade-off between efficiency and effectiveness.

## C.5 BENCHMARK SELECTION AND SCALING

Our evaluation framework employs widely-adopted benchmarks in the field. For utility assessment, we use MT-bench, which has received over 2,000 citations, while for safety evaluation, we utilize AdvBench, which has been cited more than 800 times. While we acknowledge the potential scale differences between these benchmarks based on our experimental results, they represent current standard practices in the field.

Table 10: Fine-tuning aligned LLMs on a few (10, 50, 100) harmful examples for 5 epochs. For safety evaluation, we show the ASR(%) for each fine-tuned model. For utility evaluation, we show the MT-bench score and AlpacaEval of the model after being finetuned with 100 harmful examples.

| Model | Method | Type | Size(GB) | 10-shot | 50-shot | 100-shot | MT-bench | AlpacaEval |
|---|---|---|---|---|---|---|---|---|
| Llama-2 -7b-chat | Initial | FP16 | 12.6 | 50.00 | 80.30 | 80.00 | 6.67 | 71.37 |
| | AQLM | INT4 | 2.8 | $77.40^{\uparrow 33.20}$ | $80.50^{\uparrow 0.20}$ | $81.90^{\uparrow 1.90}$ | $6.50_{\downarrow 0.17}$ | $66.42_{\downarrow 4.95}$ |
| | | INT8 | 6.0 | $75.30^{\uparrow 25.30}$ | $78.40_{\downarrow 1.90}$ | $80.00_{\downarrow -}$ | $6.54_{\downarrow 0.13}$ | $68.85_{\downarrow 2.52}$ |
| | LLM-QAT | INT4 | 3.5 | $71.2^{\uparrow 21.20}$ | $93.8^{\uparrow 13.50}$ | $92.6^{\uparrow 12.60}$ | $6.52_{\downarrow 0.15}$ | $66.54_{\downarrow 4.83}$ |
| | | INT8 | 6.5 | $65.40^{\uparrow 15.40}$ | $88.30^{\uparrow 8.00}$ | $87.20^{\uparrow 7.20}$ | $6.58_{\downarrow 0.09}$ | $69.47_{\downarrow 1.90}$ |
| | QLoRA | INT4 | 2.8 | $85.30^{\uparrow 35.30}$ | $94.20^{\uparrow 13.90}$ | $95.70^{\uparrow 15.70}$ | $6.40_{\downarrow 0.27}$ | $63.92_{\downarrow 7.45}$ |
| | | INT8 | 6.0 | $83.20^{\uparrow 33.20}$ | $90.40^{\uparrow 10.10}$ | $92.10^{\uparrow 12.10}$ | $6.40_{\downarrow 0.27}$ | $64.05_{\downarrow 7.32}$ |
| | Q-resafe | INT4 | 3.5 | $13.50_{\downarrow 36.50}$ | $14.10_{\downarrow 66.20}$ | $13.90_{\downarrow 66.10}$ | $6.59_{\downarrow 0.08}$ | $68.51_{\downarrow 2.86}$ |
| | | INT8 | 6.5 | $12.10_{\downarrow 37.90}$ | $12.60_{\downarrow 67.70}$ | $13.20_{\downarrow 66.80}$ | $6.61_{\downarrow 0.06}$ | $70.93_{\downarrow 0.44}$ |
| Gemma-7b -instruct | Initial | FP16 | 17.1 | 42.30 | 68.90 | 70.0 | 6.25 | 66.53 |
| | AQLM | INT4 | 2.8 | $55.40^{\uparrow 13.10}$ | $65.70_{\downarrow 3.20}$ | $66.00_{\downarrow 4.00}$ | $6.10_{\downarrow 0.15}$ | $61.75_{\downarrow 4.78}$ |
| | | INT8 | 6.0 | $53.80^{\uparrow 11.50}$ | $61.60_{\downarrow 7.30}$ | $63.40_{\downarrow 6.60}$ | $6.20_{\downarrow 0.05}$ | $63.59_{\downarrow 2.94}$ |
| | LLM-QAT | INT4 | 3.5 | $52.90^{\uparrow 10.60}$ | $74.20^{\uparrow 5.30}$ | $75.90^{\uparrow 5.90}$ | $6.19_{\downarrow 0.06}$ | $62.85_{\downarrow 3.68}$ |
| | | INT8 | 6.5 | $50.10^{\uparrow 7.80}$ | $73.50^{\uparrow 4.60}$ | $74.3^{\uparrow 4.30}$ | $6.24_{\downarrow 0.01}$ | $64.12_{\downarrow 2.41}$ |
| | QLoRA | INT4 | 2.8 | $61.30^{\uparrow 19.00}$ | $70.70^{\uparrow 1.80}$ | $70.90^{\uparrow 0.90}$ | $6.05_{\downarrow 0.20}$ | $59.13_{\downarrow 7.40}$ |
| | | INT8 | 6.0 | $58.90^{\uparrow 16.60}$ | $70.60^{\uparrow 1.70}$ | $68.50_{\downarrow 1.50}$ | $6.11_{\downarrow 0.14}$ | $62.50_{\downarrow 4.03}$ |
| | Q-resafe | INT4 | 3.5 | $10.40_{\downarrow 31.90}$ | $10.70_{\downarrow 58.20}$ | $11.00_{\downarrow 59.00}$ | $6.21_{\downarrow 0.04}$ | $63.77_{\downarrow 2.76}$ |
| | | INT8 | 6.5 | $9.80_{\downarrow 32.50}$ | $10.30_{\downarrow 58.60}$ | $10.70_{\downarrow 59.30}$ | $6.24_{\downarrow 0.01}$ | $66.10_{\downarrow 0.43}$ |

