# OpenReview forum: "Q-resafe: Assessing Safety Risks and Quantization-aware Safety Patching for Quantized Large Language Models"
_ICLR.cc/2025/Conference — ICLR 2025 Conference Withdrawn Submission_

### Official Review · Reviewer_6mMQ · 2024-10-26

**Soundness:** 3
**Presentation:** 2
**Contribution:** 2
**Rating:** 5
**Confidence:** 3

**Summary:**

The paper aims to assess the affect on safety alignment of quantization methods, and propose a safety patching method, Q-safe to improve the safety alignment of quantized models. The paper reveals a phenomenon that safety alignment is more damaged than general utility after quantization, considered different models and quantization methods. Q-safe mitigates this degradation by fine-tuning the quantizied model.

**Strengths:**

* The problem the paper looks at is interesting. Quantization methods indead need to ensure safety alignment as well.
* The experiment reveals an interesting phenomenon that quantization damages safety more than utility.

**Weaknesses:**

* The experiment setup (Section 3: Assessing safety risks of quantization) is confusing in some places. It considers the cases where using harmful datasets for quantization, which does not sound realistic. The used datasets for quantization are small in size.
* The experiment of the safety patching method on quantizied models does not include baselines or ablation studies.
* It is unclear about techical contributions of Q-safe as its methodology looks similar to general safety alignment methods, instead of focusing on the quantization models.

**Questions:**

Thanks for submitting the paper. I agree with the motivation of the paper that quantization needs to preserve safety enforcement as well, which is not explicitly analyzed in existing work. The paper made a contribution in this domain. However, problems in experiment setups and technical contributions make me unsure if the current version is ready to publish.

First of all, I am convined that safety is affected after quantization given the empirical analysis Section 3. Based on the phenomenon, I am curious if the authors have hypothesis or insights why safety is affected more than utility. Also, the experiments of safety assessment are confusing in some places:

The metrics are critical for understanding multiple tables but details are placed in Appenfix Table 5. For instance, Table 3 says the safety metric is $ASR_{Vanilla}$ but according to Table 5, the safety metric corresponding to quantization with Indirect Harmful Datasets should be $ASR_{AoA}$, and for AWQ the metric should be $ASR_{Decoding}$. Please be precise in using the notations.

Quantization datasets and the dataset for safety assessment all rely on AdvBench. Is there potential risk of overlap of training/fine-tuning and testing data?

In Table 5, the harmful datasets for quantization sounds quite small. As far as I see, fine-tuning of LLM quantification requires large amount of data, such as hundreds of gigabytes in the original LLM-QAT work. Also, in terms of harmful datasets for quantization, do you mean adding additional harmful data to benign datasets, or do quantization with completely harmful datasets? It does not sound realistic if it is the later. Also, what if involving safety-enforced conversations in the datasets for quantization?

Second, intuitively I think Q-safe, the safety patching method after quantization is valid in its design. However, the experiments lack a comprehensive analysis of the characteristic of the method. I have following questions:

What are the baselines, or ablation study? Baselines in my mind include: (1) directly using dataset for safety fine-tuning during the quantization; (2) directly using dataset for safety fine-tuning when using Q-safe, to show the benefit of the safety-patching dataset construction; (3) do simple LoRA fine-tuning when using Q-safe, to show the benefit of periodic safety-critical weights identification.

What is the technical contribution comparing with methods of safety alignment on LLMs, e.g., instruction tuning with safety-aligned data adopted by Llama (Llama: Open and efficient foundation language models)? I raised this question because the design of Q-safe looks independent with quantization and can work on any LLMs.

Minor issues:
* Section 4.1 Row 229, it is Table 5 not Appendix 5.
* Abbreviation DPO is used at the first time in Introduction, row 112, without explanation.

**Details Of Ethics Concerns:**

No ethic problems found.

---

> ### Author Response · Authors · 2024-11-23
> **Reply for question (part 1)**
>
> >Q1: Quantization datasets and the dataset for safety assessment all rely on AdvBench. Is there a potential risk of overlap of training/fine-tuning and testing data?.*
>
> We apologize for causing the confusion. Advbench contains a total of 520 harmful prompts. For our experiment, we used the first 10 prompts for the calibration dataset (to be consistent with existing practice [1]) for training/finetuning purposes, and the remaining 510 prompts for ASR evaluation, serving as the testing dataset. As a result, there is no overlap between the training/finetuning and testing data. We have revised the description regarding Advbench in the experimental settings (c.f. Appendix C.2) to reflect this more accurately.
>
> >Q2: Regarding harmful datasets for quantization.*
>
> We appreciate your detailed question regarding the size and composition of the harmful datasets used for quantization. We set the harmful dataset size to 10 mainly to be consistent with existing safety evaluation practices utilized for full-precision LLMs, which revealed that merely 10 harmful samples can severely degrade the safety capabilities of fine-tuned LLMs. In our supplementary experiments, harmful datasets were mixed with benign datasets to create a Mixed dataset (Ultrachat_200k + 1k harmful instructions), mimicing a balance between harmful and benign instructions in real-world datasets.
>
> For example, the results for Llama-2-7b-chat 4bit-quantization with this Mixed dataset, fine-tuned for 1 epoch, are as follows:
>
> | **Llama-2-7b-chat** | **ASR** | **MT-bench score (Utility)** |
> | ------------------- | ------- | ---------------------------- |
> | AQLM                | 20.3    | 7.1                          |
> | LLM-QAT             | 19.4    | 7.1                          |
> | QLoRA               | 48.1    | 7.0                          |
> | Q-resafe            | 5.5     | 7.1                          |
>
> This setup allows us to evaluate safety risks in a more comprehensive manner while maintaining a practical data scale. We agree that incorporating safety-enforced conversations into the datasets could provide additional insights, and we have discussed it in the future work of the updated paper. Thank you for your valuable suggestion.
>
>
> >Q3: Ablation study*
>
> Thank you for your suggestion. We have conducted the following ablation experiment based on the baseline you provided.
>
> >Q3-1: Impact of locating safety-critical weights in LoRA fine-tuning*
>
> We design the safety-critical weights locating step in Q-resafe to improve the efficiency of safety patching while restoring the safety of the quantized LLMs to a satisfactory degree. This is motivated by the recent discovery that safety-critical regions are very sparse in aligned LLMs [1]. Our experiments have shown that our locating step can indeed significantly improve the efficiency of safety-patching. As you suggested, to further demonstrate the impact of safety-critical positioning, we conducted an additional ablation experiment by evaluating a series of percentages $\tau$ of the located safety-critical weights, we select the top-$\tau$ of weights in terms of safety-criticalness for updating during the safety-patching process. For example, $\tau = 1$ means updating all weights during safety patching (no locating step), while $\tau = 0.2$ means locating $20\%$ weights to be updated. The results are reported in the table below, which is conducted on Llama-2-7b-chat for 4-bit quantization using a benign dataset (Ultrachat_200k) for 1 epoch on four A100s. The results show that the locating step has saved significant safety-patching time, and the percentage of located safety-critical weights can exert a trade-off between safety and efficiency during the safety-patching process.
>
>
> | Safety-criticle Weights Percentage ($\tau$) | ASR (Safety) | Tuning Time (Efficiency) | MT-bench score (Utility) |
> | ------------------------------------------- | ------------ | ------------------------ | ------------------------ |
> | 1                                           | 1.6          | 2.1h                     | 7.3                      |
> | 0.8                                         | 1.6          | 1.8h                     | 7.2                      |
> | 0.6                                         | 1.8          | 1.2h                     | 7.1                      |
> | 0.4                                         | 5.5          | 0.8h                     | 6.8                      |
> | 0.2                                         | 13.9         | 0.5h                     | 6.6                      |
> | 0                                           | 42.2         | -                        | 6.4                      |
>
>
>
> Reference:
>
> [1] Qi, Xiangyu, et al. "Fine-tuning Aligned Language Models Compromises Safety, Even When Users Do Not Intend To!" ICLR 2024.

---

> ### Author Response · Authors · 2024-11-23
> **Reply for question (part 2)**
>
> >Q3-2: Comparison with other baselines*
> Due to the limited time during the rebuttal period, we plan to further incorporate the following comparisons afterward:
> 1) Directly using the dataset for safety fine-tuning during quantization; 2) Directly using the dataset for safety fine-tuning when using Q-resafe, to show the benefit of the safety-patching dataset construction.
>
> >Q4: What is the technical contribution comparing with methods of safety alignment on LLMs, e.g., instruction tuning with safety-aligned data adopted by Llama (Llama: Open and efficient foundation language models)? I raised this question because the design of Q-safe looks independent with quantization and can work on any LLMs.*
>
> Thank you for raising this question. Q-resafe differs from instruction tuning methods, such as those used in Llama, in several ways. While instruction tuning aligns models during training by leveraging safety-specific data, Q-resafe focuses on addressing safety degradation in **quantized LLMs**. Specifically, Q-resafe provides: (1) **Dynamic Safety-Patching**: Q-resafe recalibrates safety-critical weights dynamically during deployment, ensuring adaptability to evolving inputs and datasets without requiring retraining. (2) **Efficiency in Safety Alignment**: By identifying sparse safety-critical regions in quantized LLMs, Q-resafe ensures efficient fine-tuning with minimal computational overhead, making it highly practical for quantized LLMs. (3) **Broad Applicability**: Q-resafe can be used to restore the safety capabilities of quantized LLMs obtained through various quantization techniques, including both two types of QAT approaches and two types of PTQ approaches. While Q-resafe can be used with any LLM, its primary focus is mitigating the specific safety risks introduced by quantization, which distinguishes it from general safety alignment methods like instruction tuning.
>
> We hope this comparison clarifies the unique contributions of Q-resafe and its complementary role alongside other safety alignment approaches.

---

> > ### Comment · Reviewer_6mMQ · 2024-11-25
> > **Response to the rebuttal**
> >
> > Thannks for submitting the rebuttal. I acknowledged and reviewed the revised version.

---

> > > ### Author Response · Authors · 2024-11-27
> > > **Grateful for Your Feedback – Revised Manuscript with Refined Safety Evaluation Claims and Enhanced Literature Review Submitted**
> > >
> > > Thank you for your prompt response and for taking the time to review our revision plan. We truly value your insightful feedback and constructive suggestions. In particular, we appreciate your guidance on avoiding overly bold claims that could potentially cause confusion within the community. We have carefully considered your recommendations and made the necessary revisions to address the concerns you raised. The updated PDF now includes more moderate and substantiated claims regarding the contributions of our safety evaluations, in line with your suggestions. Specifically:
> > >
> > > > **In the Abstract, we made the following modifications:**
> > >
> > > 1. Highlighted that existing safety studies on quantized LLMs already revealed the safety concerns.
> > > 2. Framed our safety evaluation as a complementary contribution to these existing efforts.
> > > 3. Substantiated our contribution by specifying that our study covers four mainstream quantization techniques across diverse settings, including varying quantization bit-widths and different quantization-assisting datasets.
> > >
> > > > **In Section 1 Introduction, we made the following modifications:**
> > >
> > > 1. Removed descriptions suggesting that safety evaluation is underexplored.
> > > 2. Highlighted that our safety evaluation on quantized LLMs complements existing studies.
> > > 3. Substantiated the contributions of our safety evaluation by specifying that our study addresses four mainstream quantization techniques across diverse settings, including varying quantization bit-widths and different quantization-assisting datasets.
> > >
> > > > **In Section 2 Related Works, we added a new subsection (Section 2.3) to discuss existing research that pioneered the study on the safety issues of quantized LLMs in detail.**
> > >
> > >
> > >
> > > > **In Section 3, we made the following additions and clarifications:**
> > >
> > > 1.We added references and clarifying statements in lines 204-209 to avoid confusion. We reorganized the datasets into categories (direct harmful, indirect harmful, and benign) and classified them based on risk levels to better reflect real-world applications.
> > >
> > > 2.We included relevant literature in line 413, providing further context on how our work identifies safety-critical weights.
> > >
> > > > **Futhermore, we have included new experiment results and clarificaitons requested by all reviewers.**
> > >
> > > 1.We conducted an experiment using the Llama-2-7b-chat model with a benign dataset (Ultrachat) for one epoch, extending our safety analysis of quantized models to multiple bit-widths (8/4/3/2 bit-widths).
> > >
> > > 2.We conducted another experiment with the Llama-2-7b-chat model and the same benign dataset (Ultrachat) for one epoch, analyzing the impact of identifying safety-critical weights and performing an ablation study.
> > >
> > > We hope these modifications adequately address your concerns. We greatly appreciate your constructive suggestions, which have significantly improved the quality of our work. Additionally, if you have any further suggestions, please feel free to let us know, and we would be happy to address them.
> > >
> > > Thank you again for your time and efforts.

---

> > > > ### Comment · Reviewer_6mMQ · 2024-11-27
> > > > **Feedbeck on the revised version**
> > > >
> > > > Thank for the reminder. I decided to raise my score but I am still slightly negative on it because of the following issues.
> > > >
> > > > By reading the previous work Qi, Xiangyu, et al [1], I now understand why authors design the safety assessment by involving unsafe prompts. For non-expert readers like me, I hope the author explain more about the risk level settings and their implication in the real applications (Section 3.1). The related work Qi, Xiangyu, et al [1] somehow damages the novelty of safety assessment on model quantization but I am glad the author discuss the work explicitly.
> > > >
> > > > The main concern of mine is the alignment of the problem and the solution (my Q4). Obviously, fine-tuning on safety-critical parameters should provide efficiency and flexibility in general, these methods are not new and not specialized for model quantization. So I suggest the author to finish ablation study (Q3-2) to show the benefit of each design component. Author's response to Q3-1 did a good job on this issue.
> > > >
> > > > The writing can be further improved. For instance, In section 5.2 `Safety-patching results on benign datasets` is quite vague about how this benign dataset is used. Both methdology and evaluation have lots of parameters and settings, but it is fairly hard to get this detail from the current version. I recommend the authors to have a formal definition or table to clarify.
> > > >
> > > > Thanks.

---

> > > > > ### Author Response · Authors · 2024-11-29
> > > > > **Grateful for Your Feedback - Added the Results of the Q3-2 Ablation Study**
> > > > >
> > > > > We sincerely appreciate the reviewer’s deep understanding and constructive feedback on our work. We are also grateful to the committee for extending the discussion period, which has allowed us to incorporate additional experimental results and revise the manuscript. These revisions further clarify how we designed and implemented different risk levels for testing datasets in the safety analysis of quantized models, tailored to real-world application scenarios. This dataset design enables us to better reflect the safety performance of quantized models across various contexts.
> > > > >
> > > > >
> > > > >
> > > > > > Support for Fine-Tuning Safety-Critical Weights and Efficiency Demonstration (Q3-2 Ablation Study)
> > > > >
> > > > > We fully agree with the reviewer’s suggestion regarding fine-tuning safety-critical parameters. We believe that fine-tuning should balance both flexibility and efficiency. To this end, we have conducted new experiments to further analyze the efficiency of performing safety patching under the Q-resafe framework.
> > > > >
> > > > > Building upon the previously discussed experiments on identifying safety-critical weights, we carried out three important ablation studies:
> > > > >
> > > > > 1. Directly fine-tuning using the dataset during quantization to evaluate its impact on safety.
> > > > > 2. Fine-tuning using Q-resafe with the same dataset, to demonstrate Q-resafe’s effectiveness in restoring safety.
> > > > > 3. Applying LoRA fine-tuning with Q-resafe, to illustrate the benefits of periodically identifying safety-critical weights in balancing efficiency and safety.
> > > > >
> > > > >
> > > > > Specifically, following the guidance of [1], we conducted experiments on the Llama-2-7b-chat model using the Alpaca dataset as the safety-alignment dataset, training for two epochs. Since our safety patching method integrates DPO and safety-critical parameter updates, our approach is not only safer than standard SFT methods but also more efficient than DPO with full parameter updates.
> > > > >
> > > > >
> > > > >
> > > > > The table below summarizes the results of our experiments:
> > > > >
> > > > > | Methods                              | ASR  | Training Time |
> > > > > | ------------------------------------ | ---- | ------------- |
> > > > > | LLM-QAT + SFT                        | 12.4 | 8.4h          |
> > > > > | LLM-QAT + DPO                        | 1.5  | 9.6h          |
> > > > > | LLM-QAT + Safety Patch ($\tau=0.6$)  | 1.6  | 1.2h          |
> > > > > | QLoRA + SFT                          | 26.9 | 3.4h          |
> > > > > | QLoRA + DPO                          | 2.4  | 3.8h          |
> > > > > | Q-resafe + Safety Patch ($\tau=0.6$) | 2.4  | 1.2h          |
> > > > >
> > > > >
> > > > >
> > > > > We are deeply grateful for your thoughtful feedback, which has greatly contributed to improving the rigor and depth of our work. We hope that these revisions and additional results effectively address your concerns. Should you have any further questions or suggestions, we would be delighted to discuss them and explore additional ways to refine our study. Thank you again for your insightful comments and your engagement with our research.
> > > > >
> > > > >
> > > > >
> > > > > Reference:
> > > > >
> > > > > [1]https://github.com/PKU-Alignment/safe-rlhf.git

---

> ### Author Response · Authors · 2024-12-02
> **Kindly Reminder: Rebuttal Period Ending Soon**
>
> Dear Reviewer 6mMQ,
>
> We sincerely appreciate the time and effort you have devoted to reviewing our work. We hope that our responses and revisions have addressed your concerns effectively.
>
> As part of our revisions, we have also included additional safety evaluation results to provide a more comprehensive perspective on the safety performance of quantized models. If you are considering revising your score, we would be deeply grateful for your support. Should you have any further questions or suggestions, please do not hesitate to let us know — we would be more than happy to continue the discussion and further improve our work.
>
> Thank you once again for your invaluable feedback and thoughtful engagement.
>
> Best regards,
>
> The author of paper 6792

---

### Official Review · Reviewer_qMoU · 2024-10-28

**Soundness:** 3
**Presentation:** 3
**Contribution:** 2
**Rating:** 5
**Confidence:** 3

**Summary:**

With weights quantized into smaller bit-widths, LLMs are losing performance in utility. This paper observes LLM quantization from a safety perspective and conducts a measurement study concerning the impacts of quantization on LLM safety performance. The measurement study considers varying quantization bit-widths, different quantization strategies, and different settings of quantization-assisting datasets. It reveals that quantization can significantly comprise the safety ability of the quantized LLMs.

To mitigate the safety degradation, this work proposes to patch the quantized LLM in a similar fashion to the LoRA adapter. They construct a safety-related preference dataset by collecting the responses from the full-precision LLM and the quantized LLM. Employing the preference pairs, they use the DPO loss to implement a safety alignment. Concretely, they periodically locate safety-critical weights and focusing updating them to restore the safety ability. The experiments cover three preference dataset settings and yield positive results with the Q-resafe framework.

**Strengths:**

1. This paper raises an intriguing research question: how does quantization impact LLM safety performance? To answer this, the work makes a comprehensive measurement across different quantization settings.

2. The proposed Q-resafe method is effective. It can almost perfectly sustain the safety ability of quantized LLM in the studied safety evaluation.

**Weaknesses:**

1. **Intriguing but less novel topic.** For the safety risks coming with quantization, there have been a great number of works revealing it, from the vision field (e.g., https://dl.acm.org/doi/10.1145/3485832.3485881) to recent LLM research (e.g., https://arxiv.org/abs/2405.18137). From another perspective, safety performance is just one intrinsic ability of LLM. The finding of safety degradation after quantization is not impressive.
2. **Writing is not self-inclusive.** The four considered quantization methods are essential for understanding why they behave differently. The lack of technical details of the surveyed methods hinders understanding the rationales behind the security risks incurred by quantization.
3. **Lack of ablation study.** I have not noticed why the operation of safety-critical weight identification is necessary. And no ablation study provides evidence for employing it.
4. **Lack of impressive insights**: To be honest, I think the analysis for the measurement results is not in-depth. More discussions about safety benchmarks and utility benchmarks are in demand. Although it looks scary in Table 3 that the safety performance is almost compromised after quantization, utility and safety performance may be non-equally scaled.

Minor mistakes:
1. Mistaken references are observed in line 190 and line 402.
2. The symbol $Q_0$ in Eq. 2 I think should be $Q^0$.
3. In line 413, `As recent studies have observed, LLMs exhibit localization properties.` What recent studies it refers to?
4. Please use the same title in the paper and the website submission.

**Questions:**

- Can we collect the safety-related preference dataset in Q-resafe for offline use? For example, to substantiate an auxiliary loss to guide the safety-in-mind QAT.
- The design of locating safety-critical weights assumes that the safety ability of LLMs can located in a limited pool of weights. It is unclear whether they exhibit a positive or negative relationship with safety? If positive, what about sustaining the precision of the safety-critical weights and only quantizing other weights? From a safety-at-first perspective, how strong safety performance can such a method ensure?
- Safety is a vague concept. Without data, how to identify the security-critical weights? Are detailed descriptions missing?
- Is the safety ability of the quantized LLM always bottlenecked by its full-precision counterpart?

---

> ### Author Response · Authors · 2024-11-23
> **Reply for weaknesses**
>
> >W1 Regarding novelty
>
> We appreciate the reviewer’s observation regarding the novelty of our work. While we acknowledge that safety issues in quantization have been explored in other domains, such as vision, and some aspects in LLMs, our work aims to address a key gap: systematically evaluating and mitigating safety degradation specifically in quantized LLMs. Existing studies primarily focus on accuracy and utility during quantization, while our work emphasizes safety-utility trade-offs, introducing a dynamic safety-patching mechanism to address these challenges. We have revised the related work section to clarify these distinctions and better position our contributions within the existing body of research.
>
> >W2 Regarding self-inclusive
>
> We agree that the explanation of surveyed quantization methods could benefit from more technical details. In response, we will expand the descriptions of the quantization techniques (e.g., AWQ, QLoRA) in the revised manuscript, particularly focusing on their safety and utility trade-offs. We will also provide further insights into why certain methods perform differently under safety evaluations. This revision will help readers better understand the rationale behind their behavior.
>
> >W3&Q2 Regarding impact of locating safety-critical weights* and its ablation study
>
> We design the safety-critical weights locating step in Q-resafe to improve the efficiency of safety patching while restoring the safety of the quantized LLMs to a satisfactory degree. This is motivated by the recent discovery that safety-critical regions are very sparse in aligned LLMs [1]. Our experiments have shown that our locating step can indeed significantly improve the efficiency of safety-patching. As you suggested, to further demonstrate the impact of safety-critical positioning, we conducted an additional ablation experiment by evaluating a series of percentages $\tau$ of the located safety-critical weights, we select the top-$\tau$ of weights in terms of safety-criticalness for updating during the safety-patching process. For example, $\tau = 1$ means updating all weights during safety patching (no locating step), while $\tau = 0.2$ means locating $20\%$ weights to be updated. The results are reported in the table below, which is conducted on Llama-2-7b-chat for 4-bit quantization using a benign dataset (Ultrachat_200k) for 1 epoch on four A100s. The results show that the locating step has saved significant safety-patching time, and the percentage of located safety-critical weights can exert a trade-off between safety and efficiency during the safety-patching process.
>
>
> | Safety threshold ($\tau$) | ASR (Safety) | Safety-patching Time (Efficiency) | MT-bench score (Utility) |
> | ------------------------- | ------------ | --------------------------------- | ------------------------ |
> | 1                         | 1.6          | 2.1h                              | 7.3                      |
> | 0.8                       | 1.6          | 1.8h                              | 7.2                      |
> | 0.6                       | 1.8          | 1.2h                              | 7.1                      |
> | 0.4                       | 5.5          | 0.8h                              | 6.8                      |
> | 0.2                       | 13.9         | 0.5h                              | 6.6                      |
> | 0                         | 42.2         | -                                 | 6.4                      |
>
>
> >W4 Regarding "more discussions about safety benchmarks and utility benchmarks" and "utility and safety performance may be non-equally scaled"
>
> We have adopted the most common testing method. The MT-bench for utiltiy evaluation has been cited over 2,000 times, and the Advbench for safety evaluation has been cited over 800 times. We understand that your concern is that the scale of MT-bench is smaller based on the experimental results. We will add more benchmarks such as Alpaca Winrate and perplexity in the future.

---

> ### Author Response · Authors · 2024-11-23
> **Reply for Questions**
>
> > Q1: Can we collect the safety-related preference dataset in Q-resafe for offline use? For example, to substantiate an auxiliary loss to guide the safety-in-mind QAT.
>
> We agree that developing a safety-in-mind QAT approach is a promising alternative, as you suggested. In our paper, we opted to develop a post-hoc safety-patching method to restore the safety capabilities of an already-quantized LLM, primarily due to its broad applicability. That is, our Q-resafe technique can be used to restore the safety capabilities of quantized LLMs obtained through various quantization techniques, including both QAT and PTQ. We believe that Q-resafe has the potential to be repurposed for for safety-in-mind QAT by incorporating its loss function as a safety regularization term and restricting this regularization effect to the identified safety-critical weights to avoid interfering with the utility-preserving purpose. We have discussed this potential application as future work in the updated paper.
>
> >Q3: Safety is a vague concept. Without data, how to identify the security-critical weights? Are detailed descriptions missing?*
>
> We apologize for causing the confusion. We identify the safety-critical weights with SNIP score [1-2]. In detail, for a prompt $x$ and response $y$, we take the loss as the conditional negative log-likelihood $\mathcal{L}(x)=-\log p(y|x)$ predicted by the model. For any layer with weight matrix $W$, the importance score for each weight entry $W_{ij}$ as $$I(W_{ij},x)=|W_{ij} \cdot \nabla_{W_{ij}}\mathcal{L}(x)|$$, Given a calibration dataset $D$, we take the average value and obtain $I(W_{ij})=\mathbb{E}_{x \in D}I(W_{ij},x)$.
> The calibration dataset is critile and there is currently no way to identify safety-critical weights for LLMs without a calibration set, just as you suggested. The calibration dataset here refers to harmful prompts, such as: “How to make a bomb?“. In our paper, it suffices to use 5-10 harmful prompts to accurately identify the location of safety-critical weights, and generating these harmful prompts is effortless, e.g., by manually written by human {USER: Write a script to attack the website; ASSISTANT: Sure, here is the script... }
>
> >Q4: Is the safety ability of the quantized LLM always bottlenecked by its full-precision counterpart?*
>
> We opted for a self-supervised approach for the safety-patching algorithm, where we use the full-precision counterpart to provide safety guidance for the quantized LLM. Alternatively, it is also possible to utilize external LLMs, such as Llama-chat and Gemma-Instruct, to provide the safety guidance. This approach has the potential to further enhance the safety capabilities of the quantized LLM. We have added a new discussion about utilizing external LLMs for safety guidance in the updated PDF. Thank you for your suggestion.
>
> >Minor mistakes *
>
> Thank you for pointing out these minor mistakes. We have thoroughly addressed them in the updated PDF.
>
>
> Referencens：
>
> [1] Wei, Boyi, et al. "Assessing the brittleness of safety alignment via pruning and low-rank modifications." ICML'24.
>
> [2] Xie, Yueqi, et al. "GradSafe: Detecting Jailbreak Prompts for LLMs via Safety-Critical Gradient Analysis." ACL'24.

---

> > ### Comment · Reviewer_qMoU · 2024-11-26
> >
> > Thanks for the clarification. The responses address most of my concerns.
> >
> > However, I am still concerned about the representativeness of one single security benchmark. Due to the potential out-of-alignment scales of different benchmarks, the severity of the problem resulting from quantization is still unclear.
> >
> > I decided to raise my score, but I still hold a negative point about the contribution of the work.

---

> > > ### Author Response · Authors · 2024-11-26
> > > **Response to Reviewer’s Concerns about Safety Benchmarks**
> > >
> > > Thank you for your valuable feedback regarding the representativeness of safety benchmarks and the depth of the analysis. Below, we would like to provide further feedback to more adequately address your concerns.
> > >
> > > > Additional experiments beyond AdvBench
> > >
> > > To complement AdvBench, we conducted an additional safety evaluation using a **harmfulness scoring metric**. This evaluation case includes 330 examples across 11 categories of prohibited use cases based on the **OpenAI Usage Policy** and **Llama-2 Usage Policy**, with 30 examples per category. For evaluation, we exploit GPT-4 as the judge by providing it with a prompt containing: The prohibited use case policy, the harmful instruction, the model's output for this instruction, a detailed rating rubric, which follows existing works on safety evaluations [3,4]. Each (harmful instruction, model response) pair is assigned a **harmful score** ranging from 1 to 5, where a higher score indicates a greater level of harmfulness. We report the **average harmful score** (HS) across all instructions as well as the harmful rate, defined as the percentage of test cases that received the maximum harmfulness score of 5. To minimize randomness, we set the temperature $\tau$ and $top-p$ parameters to 0 during inference.
> > >
> > > The table below presents harmful score results on a benign dataset (Ultrachat) with 100 harmful examples across different quantization methods and precision levels, offering an additional perspective on safety degradation in quantized LLMs. We hope that the new results will help address your concern about relying on a single benchmark. We will incorporate these findings into the revised manuscript.
> > >
> > > | **Model**             | **Method** | **Type** | **Harmful Score (HS)** |
> > > | --------------------- | ---------- | -------- | ---------------------- |
> > > | **Llama-2-7b-chat**   | Initial    | FP16     | 1.00                   |
> > > |                       | AQLM       | INT4     | 1.97                   |
> > > |                       |            | INT8     | 1.89                   |
> > > |                       | LLM-QAT    | INT4     | 1.90                   |
> > > |                       |            | INT8     | 1.82                   |
> > > |                       | QLoRA      | INT4     | 3.01                   |
> > > |                       |            | INT8     | 2.97                   |
> > > |                       | Q-resafe   | INT4     | 1.25                   |
> > > |                       |            | INT8     | 1.24                   |
> > > | **Gemma-7b-instruct** | Initial    | FP16     | 9.20                   |
> > > |                       | AQLM       | INT4     | 2.26                   |
> > > |                       |            | INT8     | 2.20                   |
> > > |                       | LLM-QAT    | INT4     | 2.07                   |
> > > |                       |            | INT8     | 1.97                   |
> > > |                       | QLoRA      | INT4     | 2.85                   |
> > > |                       |            | INT8     | 2.78                   |
> > > |                       | Q-resafe   | INT4     | 1.61                   |
> > > |                       |            | INT8     | 1.60                   |
> > >
> > >
> > >
> > > > Reason why we chose Advbench in submission
> > >
> > > AdvBench remains a widely accepted benchmark in the field and has been utilized in many recent works [5,6,7,8] to evaluate safety, where it has proven sufficient to reflect key safety risks in LLMs. In order to be consistent and allow for direct comparison with these works, we primarily considered AdvBench in the submitted paper. While we acknowledge that no single benchmark can capture all aspects of safety, we believe AdvBench provides a solid foundation for evaluating the safety impact of quantization.
> > >
> > > > Future Work and Integration of Additional Benchmarks
> > >
> > > Although AdvBench provides a solid foundation for evaluating the safety impact of quantization, we agree with you that no single benchmark can capture all aspects of safety and that a more comprehensive safety evaluation framework would further strengthen this work. Following your suggestion, we have provided new safety evaluation results above. We are willing to consider more safety benchmarks, such as TruthfulQA and HarmlevelBench, or any other safety benchmarks you may suggest. We sincerely appreciate your valuable comments, which have helped us identify areas for improvement and plan for future extensions of this work. Thank you again for your thoughtful comments.

---

> > > > ### Author Response · Authors · 2024-11-26
> > > > **Response to Reviewer’s Concerns about Safety Benchmarks with Reference**
> > > >
> > > > The following references are cited in our response to provide supporting evidence and context for the points discussed.
> > > >
> > > > **Reference:**
> > > >
> > > > [3] OpenAI. GPT-4V(ision) system card. https://openai.com/research/gpt-4v-system-card, 2023.
> > > >
> > > > [4] OpenAI. Gpt-4 technical report, 2023.
> > > >
> > > > [5] Yang X, Wang X, Zhang Q, et al. Shadow alignment: The ease of subverting safely-aligned language models[J]. arXiv preprint arXiv:2310.02949, 2023.
> > > >
> > > > [6] Kumar D, Kumar A, Agarwal S, et al. Fine-Tuning, Quantization, and LLMs: Navigating Unintended Outcomes[J]. arXiv preprint arXiv:2404.04392, 2024.
> > > >
> > > > [7] Li S, Yao L, Zhang L, et al. Safety Layers in Aligned Large Language Models: The Key to LLM Security[J]. arXiv preprint arXiv:2408.17003, 2024.
> > > >
> > > > [8] Qi X, Zeng Y, Xie T, et al. Fine-tuning aligned language models compromises safety, even when users do not intend to![J]. arXiv preprint arXiv:2310.03693, 2023.

---

> > > ### Author Response · Authors · 2024-11-28
> > > **Grateful for Your Feedback – Revised Manuscript with Refined Safety Evaluation Claims and Enhanced Literature Review Submitted**
> > >
> > > Dear Reviewer,
> > >
> > > We sincerely appreciate your prompt response and insightful comments on our manuscript. Your constructive feedback has been invaluable, and we are grateful for your careful attention to the details of our work. In light of your suggestions, we have revised our manuscript and uploaded an updated version of the PDF. These revisions address your concerns and ensure that our safety evaluation claims are both well-supported and appropriately moderated, as per your recommendations.Specifically:
> > >
> > > > **In the Abstract, we made the following modifications:**
> > >
> > > 1. Highlighted that existing safety studies on quantized LLMs already revealed the safety concerns.
> > > 2. Framed our safety evaluation as a complementary contribution to these existing efforts.
> > > 3. Substantiated our contribution by specifying that our study covers four mainstream quantization techniques across diverse settings, including varying quantization bit-widths and different quantization-assisting datasets.
> > >
> > > > **In Section 1 Introduction, we made the following modifications:**
> > >
> > > 1. Removed descriptions suggesting that safety evaluation is underexplored.
> > > 2. Highlighted that our safety evaluation on quantized LLMs complements existing studies.
> > > 3. Substantiated the contributions of our safety evaluation by specifying that our study addresses four mainstream quantization techniques across diverse settings, including varying quantization bit-widths and different quantization-assisting datasets.
> > > 4. We have revised lines 123 to improve clarity and apologize for any confusion caused previously.
> > >
> > > > **In Section 2 Related Works, we added a new subsection (Section 2.3) to discuss existing research that pioneered the study on the safety issues of quantized LLMs in detail.**
> > >
> > > 1.In line 256, we clarified that this section refers to the ASR values of Llama-2 after adjusting the sampling temperature. Table 2 shows the ASR values of the safety-aligned original model, and Table 3 corresponds to the configuration without fine-tuning.
> > >
> > > > **In Section 3, we made the following additions and clarifications:**
> > >
> > > 1.We included relevant literature in line 413, providing further context on how our work identifies safety-critical weights.
> > >
> > > > **In the Conclusion and Future Work sections, we have discussed the limitations of our work and made adjustments to the formatting details.**
> > >
> > > > **Futhermore, we have included new experiment results and clarificaitons requested by all reviewers.**
> > >
> > > 1.We conducted an experiment using the Llama-2-7b-chat model with a benign dataset (Ultrachat) for one epoch, extending our safety analysis of quantized models to multiple bit-widths (8/4/3/2 bit-widths).
> > >
> > > 2.We conducted another experiment with the Llama-2-7b-chat model and the same benign dataset (Ultrachat) for one epoch, analyzing the impact of identifying safety-critical weights (0/0.2/0.4/0.6/0/8/1.0) and performing an ablation study.
> > >
> > > 3.We incorporated additional widely-used quantization algorithms (LLM.int8(), NF4, and FP4), which are frequently used in practice and could pose significant safety risks to users. Our additional experimental results show that these quantization techniques also compromise the safety of quantized LLMs, and Q-resafe provides a promising solution for mitigating this issue. We believe these results significantly enhance the completeness of our study.
> > >
> > > 4.To address concerns about the representativeness and depth of our safety benchmarks, we added experiments using harm-rating metrics as an additional safety assessment. This offers a new perspective on the safety degradation of quantized LLMs, with evaluation cases based on both OpenAI's policy and Llama-2’s policy.
> > >
> > > We sincerely hope that these modifications adequately address your concerns and demonstrate the depth of our revision. We are extremely grateful for your constructive suggestions, which have significantly improved the quality and thoroughness of our work. If you have any further suggestions or concerns, please do not hesitate to let us know, and we would be more than happy to address them.
> > >
> > > Thank you once again for your time, effort, and invaluable input.
> > >
> > > Best regards,
> > > Submission [6792] Authors

---

> ### Author Response · Authors · 2024-12-02
> **Kindly Reminder: Rebuttal Period Ending Soon**
>
> Dear Reviewer qMoU,
>
> We sincerely appreciate the time and effort you have devoted to reviewing our work. We hope that our responses and revisions have addressed your concerns effectively. If you are considering revising your score, we would be deeply grateful for your support. Should you have any further questions or suggestions, please do not hesitate to let us know — we would be more than happy to continue the discussion and further improve our work.
>
> Thank you once again for your invaluable feedback and thoughtful engagement.
>
> Best regards,
>
> The author of paper 6792

---

### Official Review · Reviewer_vSR8 · 2024-11-03

**Soundness:** 3
**Presentation:** 3
**Contribution:** 3
**Rating:** 5
**Confidence:** 4

**Summary:**

This paper studies the safety risks of quantization for LLMs. It first presents a detailed evaluation on how quantization reduces LLM’s safety, covering four quantization methods, three types of quantization-assisting datasets, and two bit-widths. To address the safety issue of quantization, the paper proposes Q-resafe, an approach that fine-tunes the safety-related weights of quantized LLMs to restore safety. The experiments demonstrate the consistent benefits of Q-resafe across the board.

**Strengths:**

The paper studies an important but relatively underexplored problem. The evaluation of existing quantization approaches clearly demonstrates the safety issues of quantization and Q-resafe gives significant benefits.

**Weaknesses:**

- The paper claims to be the first systematic assessment of safety risks of quantization. However, I am aware of at least two prior papers in this direction [1, 2]. I suggest the authors conduct a more comprehensive literature review and adjust the claim.

- While the paper studies more advanced quantization algorithms, it does not cover algorithms that are already popular, such as LLM.int8(), NF4, and FP4, implemented in the bitsandbytes library. The safety issues of these popular algorithms could have a larger impact on the users.

- The paper sometimes makes unsubstantiated claims. At lines 204-209, the paper says “Following the established practice in literature” but does not provide any reference. I am also confused at this point: why would users like to use direct harmful datasets or indirectly harmful datasets for assisting quantization? Moreover, at line 413, the paper writes “As recent studies have observed” without giving any citations. I believe identifying safety-critical weights is itself a research challenge. How does Q-resafe’s approach generalize?

- Why do the authors adopt a post-hoc fixing approach for Q-resafe? Have the authors considered incorporating safety mechanisms already during quantization?

- The writing needs some adjustments. First, line 123 is an unfinished sentence. Second, the baseline numbers for the quantization methods are not clearly presented. Table 2 says 0.3 and 9.2, but line 256 suddenly says 29.8 and 9.4, which are not mentioned before.

- On page 11, the paper has a “Limitations” section. I am not sure if this is the correct place to discuss limitations. It may not align with the ICLR formatting requirements.

[1] Hong, Junyuan, et al. "Decoding Compressed Trust: Scrutinizing the Trustworthiness of Efficient LLMs Under Compression." ICML 2024.

[2] Egashira, Kazuki, et al. "Exploiting LLM Quantization." NeurIPS 2024.

**Questions:**

Please answer the points raised in the “Weakness” section.

---

> ### Author Response · Authors · 2024-11-23
> **Thank you for your valuable suggestions. We have revised the presentation in the paper based on your comments, added relevant experiments, and clarified the motivation for data search and azimuth to better respond to your comments. (w1-w3)**
>
> > W1: The paper claims to be the first systematic assessment of safety risks of quantization. However, I am aware of at least two prior papers in this direction [1,2]. I suggest the authors conduct a more comprehensive literature review and adjust the claim.
>
> Thank you for pointing this out. We apologize for any overstatement in our claims. Following your advice, we have revised the manuscript to better position our work as building on these studies. Specifically, we have: 1) Moderated the claim about the contributions to safety evaluations on quantized LLMs in the abstract and introduction; 2) Added a dedicated subsection in the related work section to introduce the prior works you pointed out.
>
> > W2: Safety issues of LLM.int8(), NF4, and FP4
>
> Thank you for your valuable suggestion. We acknowledge the importance of evaluating popular quantization algorithms, such as LLM.int8(), NF4, and FP4, as they are widely used and could have significant safety implications for users. In response to your suggestion, we have conducted additional experiments with LLM.int8() and FP4 on Llama-7b-Chat, and we have included these results in the revised manuscript in the Additional experiment settings and results can be found in Appendix C. The additional experiment results indicate that these quantization methods also degrade the safety capabilities of quantized LLMs. Furthermore, the results demonstrate that Q-resafe restores safety after patching for these quantization methods. We hope these new experiment results address your concern and enhance the completeness of our study. Thank you again for bringing this to our attention.
>
> | **Quant Method** | **ASR before safety patch** | **ASR after safety patch** |
> | ---------------- | --------------------------- | -------------------------- |
> | LLM.int8()       | 19.2                        | 5.2                        |
> | NF4              | 23.9                        | 5.5                        |
> | FP4              | 35.2                        | 6.0                        |
>
>
>
> >W3: The paper sometimes makes unsubstantiated claims. At lines 204-209, the paper says “Following the established practice in literature” but does not provide any reference. I am also confused at this point: why would users like to use direct harmful datasets or indirectly harmful datasets for assisting quantization? Moreover, at line 413, the paper writes “As recent studies have observed” without giving any citations. I believe identifying safety-critical weights is itself a research challenge. How does Q-resafe’s approach generalize?
>
>
>
> Thank you for your valuable and detailed suggestions. Following your advice, we have addressed the specific points raised regarding unsubstantiated claims：
>
> (1) **Line 204-209 (“Following the established practice in literature”):**
> We sincerely apologize for the lack of citations in this section, which may have caused confusion. The statement refers to the reality that even seemingly benign prompts can sometimes lead to unsafe outputs from large language models. To reflect this, we organized our dataset to include **directly harmful**, **indirectly harmful**, and **benign** prompts, categorized based on risk levels. These categories follow practices outlined in prior studies, such as[3].
>
> (2) We acknowledge that this statement was missing appropriate references, and we apologize for the confusion caused. The missing citation is [4].
>
> These study highlights the risks associated with quantization, particularly in safety-critical scenarios. We have corrected the manuscript to include these studies and rephrased the statement for clarity.
>
> The motivation for including harmful datasets in our experiments stems from the observation that seemingly benign prompts can unintentionally elicit unsafe responses in large language models. By incorporating direct and indirectly harmful datasets, we aim to comprehensively assess and mitigate safety risks across a wide range of scenarios, reflecting real-world use cases. This approach ensures that the quantized models are robust not only to typical benign inputs but also to edge cases and adversarial scenarios.

---

> ### Author Response · Authors · 2024-11-23
> **Thank you for your valuable suggestions. We have made corrections based on your comments. The following are the responses to w4-w6 and the corresponding reference.**
>
> >W4: Why do the authors adopt a post-hoc fixing approach for Q-resafe? Have the authors considered incorporating safety mechanisms already during quantization?
>
> We opted to develop a post-hoc safety-patching method to restore the safety capabilities of an already-quantized LLM, primarily due to its broad applicability. That is, our Q-resafe technique can be used to restore the safety capabilities of quantized LLMs obtained through various quantization techniques, including both two types of QAT approaches and two types of PTQ approaches. We believe that Q-resafe has the potential to be adapted for during-quantization safety enhancement. For example, we can incorporate the loss of Q-resafe into QAT as a safety regularization term and restrict this regularization effect to the identified safety-critical weights to avoid interfering with the utility-preserving purpose. We have discussed this potential application as future work in the updated paper. Thank you for your insightful suggestion. **We have added a discussion of future work to the conclusion of the paper.**
>
>
>
> >W5: The writing needs some adjustments. First, line 123 is an unfinished sentence. Second, the baseline numbers for the quantization methods are not clearly presented. Table 2 says 0.3 and 9.2, but line 256 suddenly says 29.8 and 9.4, which are not mentioned before.
>
> Thank you for pointing out these issues. We sincerely apologize for the unfinished sentence on line 123.
>
> Regarding the differences in numbers between Table 2 and line 256, we would like to clarify the following:
>
> (1) **Table 2** presents ASR values for the original model after applying safety alignment.
>
> (2) **Line 256** reflects ASR values for Llama-2 with a modified sampling temperature, which differs from the original values in Table 2.
>
> In Table 3, the AWQ values correspond to configurations without fine-tuning. When evaluated across various risk-level datasets and with modified sampling strategies, the ASR values for the 4-bit and 8-bit configurations are 42.4% and 39.1%, respectively.
>
>
>
> > W6: On page 11, the paper has a “Limitations” section. I am not sure if this is the correct place to discuss limitations. It may not align with the ICLR formatting requirements.
>
>
>
> Thank you for pointing this out. We appreciate your attention to formatting details. We included the "Limitations" section on page 11 to provide a transparent discussion of our work's constraints. However, we understand that this placement may not align with ICLR formatting requirements. In response, we have relocated this section to the end of the conclusion or discussion, ensuring it adheres to the guidelines. Thank you again for your valuable feedback.
>
>
>
> Reference:
>
> [1] Lee, Namhoon, et al. "SNIP: Single-shot network pruning based on connection sensitivity." ICLR 2019.
>
> [2] Liu Zechun, et al. "Llm-qat: Data-free quantization aware training for large language models." arXiv preprint arXiv:2305.17888
>
> [3] Qi, Xiangyu, et al. "Fine-tuning Aligned Language Models Compromises Safety, Even When Users Do Not Intend To!" ICLR 2024.

---

> > ### Comment · Reviewer_vSR8 · 2024-11-25
> >
> > Thank the authors for providing the rebuttal! I have checked them but do not plan to raise my score.

---

> > > ### Author Response · Authors · 2024-11-27
> > > **Revised Manuscript with Clarifications and Updates – Addressing Your Constructive Feedback**
> > >
> > > Thank you for your prompt response and for acknowledging our revision plan. We truly appreciate your insightful comments and constructive feedback, especially your suggestion to avoid overly bold claims that could potentially lead to confusion within the community. In light of your valuable input, we have made the necessary revisions and uploaded an updated version of the PDF. This new version addresses the concerns related to the contributions of our safety evaluations, ensuring that the claims are now well-supported and appropriately moderated, as you recommended. Specifically:
> > >
> > > > **In the Abstract, we made the following modifications:**
> > >
> > > 1. Highlighted that existing safety studies on quantized LLMs already revealed the safety concerns.
> > > 2. Framed our safety evaluation as a complementary contribution to these existing efforts.
> > > 3. Substantiated our contribution by specifying that our study covers four mainstream quantization techniques across diverse settings, including varying quantization bit-widths and different quantization-assisting datasets.
> > >
> > > > **In Section 1 Introduction, we made the following modifications:**
> > >
> > > 1. Removed descriptions suggesting that safety evaluation is underexplored.
> > > 2. Highlighted that our safety evaluation on quantized LLMs complements existing studies.
> > > 3. Substantiated the contributions of our safety evaluation by specifying that our study addresses four mainstream quantization techniques across diverse settings, including varying quantization bit-widths and different quantization-assisting datasets.
> > > 4. We have revised lines 123 to improve clarity and apologize for any confusion caused previously.
> > >
> > > > **In Section 2 Related Works, we added a new subsection (Section 2.3) to discuss existing research that pioneered the study on the safety issues of quantized LLMs in detail.**
> > >
> > > 1.In line 256, we clarified that this section refers to the ASR values of Llama-2 after adjusting the sampling temperature. Table 2 shows the ASR values of the safety-aligned original model, and Table 3 corresponds to the configuration without fine-tuning.
> > >
> > > > **In Section 3, we made the following additions and clarifications:**
> > >
> > > 1.We added references and clarifying statements in lines 204-209 to avoid confusion. We reorganized the datasets into categories (direct harmful, indirect harmful, and benign) and classified them based on risk levels to better reflect real-world applications.
> > >
> > > 2.We included relevant literature in line 413, providing further context on how our work identifies safety-critical weights.
> > >
> > > > **In the Conclusion and Future Work sections, we have discussed the limitations of our work and made adjustments to the formatting details.**
> > >
> > > > **Futhermore, we have included new experiment results and clarificaitons requested by all reviewers.**
> > >
> > > 1.We conducted an experiment using the Llama-2-7b-chat model with a benign dataset (Ultrachat) for one epoch, extending our safety analysis of quantized models to multiple bit-widths (8/4/3/2 bit-widths).
> > >
> > > 2.We conducted another experiment with the Llama-2-7b-chat model and the same benign dataset (Ultrachat) for one epoch, analyzing the impact of identifying safety-critical weights and performing an ablation study.
> > >
> > > We hope these modifications adequately address your concerns. We are extremely grateful for your constructive suggestions, which have significantly improved the quality of our work. If you have any further suggestions or concerns, please do not hesitate to let us know, and we would be happy to address them.
> > >
> > > Thank you again for your time and efforts.

---

> > > ### Author Response · Authors · 2024-12-02
> > > **Kindly Reminder: Rebuttal Period Ending Soon**
> > >
> > > Dear Reviewer vSR8,
> > >
> > > We would like to express our gratitude to all the time and effort you have devoted to reviewing our work. We hope that our responses and revisions have addressed your concerns effectively. If you are considering revising your score, we would be deeply grateful for your support. Should you have any further questions or suggestions, please do not hesitate to let us know — we would be more than happy to continue the discussion and further improve our work.
> > >
> > > Thank you once again for your invaluable feedback and thoughtful engagement.
> > >
> > > Best regards,
> > >
> > > The author of paper 6792

---

### Official Review · Reviewer_Jb1g · 2024-11-06

**Soundness:** 3
**Presentation:** 3
**Contribution:** 3
**Rating:** 6
**Confidence:** 4

**Summary:**

The paper studies the extent to which the different quantization techniques degrade the safety capabilities of quantized LLMs, and how can such declines in safety capabilities be mitigated.

**Strengths:**

+ Comprehensiveness: The study covers all four mainstream categories of LLM quantization techniques covering two post-quantization techniques and two quantization-aware training/finetuning techniques. For quantization techniques needing additional quantization-assisting dataset, the paper uses three datasets with varying safety risk levels: a directly harmful dataset, an indirectly harmful dataset, and a benign dataset. The quantized LLMs are evaluated with two commonly adopted bit-widths.

+  The paper also looks at the feasibility of identifying and updating a small portion of safety-critical weights, then exploit potential tools for identifying these weights, and construct a pair of masking matrices corresponding to the LoRA variables. This is clearly a novel contribution.

**Weaknesses:**

The paper makes a claim that "Although preliminary literature reports scattered evidence of safety evaluations for quantized LLMs, they are not systematic enough to support a well-rounded evaluation". Its primarily claim to novelty appears to be that it is the first to study the safety impact of quantized LLMs. Here is a list of work studying different aspects of quantizations impact on safety and alignment:

- Belkhiter, Yannis, Giulio Zizzo, and Sergio Maffeis. "HarmLevelBench: Evaluating Harm-Level Compliance and the Impact of Quantization on Model Alignment." Neurips Safe Generative AI Workshop 2024.

- Egashira, Kazuki, Mark Vero, Robin Staab, Jingxuan He, and Martin Vechev. "Exploiting LLM Quantization." arXiv preprint arXiv:2405.18137 (2024).

- Belkhiter, Yannis, Giulio Zizzo, and Sergio Maffeis. "HarmLevelBench: Evaluating Harm-Level Compliance and the Impact of Quantization on Model Alignment." In Neurips Safe Generative AI Workshop 2024.

- Hong, Junyuan, Jinhao Duan, Chenhui Zhang, Zhangheng Li, Chulin Xie, Kelsey Lieberman, James Diffenderfer et al. "Decoding Compressed Trust: Scrutinizing the Trustworthiness of Efficient LLMs Under Compression." arXiv preprint arXiv:2403.15447 (2024).

The claim that the paper is the "first systematic safety risk assessment of quantization on LLMs and mitigate the safety degradation" is too bold and not fully substantiated. It also appears unneeded. The reviewer will raise the score if the claim is better substantiated or moderated. Letting a paper be published with such a bold not-fully-substantiated claim is not feasible as it could confuse the community.

**Questions:**

* It would be useful to study the sensitivity of ASR results with the temperature and observe if the deterioration after quantization is worse or more gradual at higher temperatures.

* "the ASR rises from 29.80% on the pre-quantization Llama-2-7b-chat model to 42.40% on the INT4 model and to 39.10% on the INT8 model. Similarly, for the Gemma-7b-instruct model, the ASR increases from 9.40% pre-quantization to 17.90% on the INT4 model and to 15.10%" - it appears that the decline is more steep going to INT8 compared to going to INT4. It would be good to add discussion on this and conduct more experiments to understand this phenomenon and see if there is any saturation.

---

> ### Author Response · Authors · 2024-11-23
> **Thank you for your valuable suggestions. We have further refined the contribution description based on your comments, added relevant work discussions, and conducted additional experiments to further analyze the impact of quantization on model security and related phenomena.**
>
> > Regarding the weakness of the claim about safety evaluations on quantized LLMs.
>
> Thank you for your valuable suggestion. We appreciate your detailed comments and agree with the valuable prior studies on the impact of quantization on safety. According to your advice, we have moderated our claim about the contribution of safety evaluation on quantized LLMs to be more specific and concrete. That is, our contribution is presenting a comprehensive safety evaluation of quantized LLMs, covering four different quantization techniques and three different quantization-assisting datasets. We have revised and emphasized this work. Furthermore, we have revised the related work section by adding a dedicated subsection {safety evaluations for LLMs} to introduce the prior works you pointed out in a more detailed and accurate manner. In addition, we would like to remark that our work builds on and complements the existing efforts by focusing on the interplay between practicality, safety, and effectiveness in quantization techniques. We propose a general safety-patching method to restore the safety capabilities of quantized LLMs, offering a versatile approach applicable across different quantization methods. We hope these revisions clarify our contributions and position our work as a meaningful addition to this growing research area.
>
> > Q1: It would be useful to study the sensitivity of ASR results with the temperature and observe if the deterioration after quantization is worse or more gradual at higher temperatures.
>
> Thank you for your insightful question regarding the sensitivity of ASR results to temperature and how quantization-induced deterioration varies across temperature levels. To address this, we conducted experiments analyzing ASR across a range of temperatures (0.6 to 0.95) for both INT4 and INT8 quantization levels, using the Llama-2-7b-Chat and Gemma-7b-Instruct models. The results are summarized in Figures 5 and 6 of Appendix.
>
> The new results show that ASR consistently increases with temperature for both models, regardless of quantization level. This behavior is attributed to higher temperatures encouraging more diverse token sampling, which can inadvertently include unsafe outputs.
>
> Additionally, we observed that each model has a specific temperature range where it achieves a better balance between safety and generation quality. For example, Vicuna and Falcon models perform optimally in the lower temperature range (0.1-0.3), while the Llama-2 series demonstrates significantly improved safety at higher temperatures (τ = 0.95). Adjusting the temperature within these ranges can mitigate quantization-induced degradation while optimizing safety and generation diversity.
>
> > Q2: "the ASR rises from 29.80% on the pre-quantization Llama-2-7b-chat model to 42.40% on the INT4 model and to 39.10% on the INT8 model. Similarly, for the Gemma-7b-instruct model, the ASR increases from 9.40% pre-quantization to 17.90% on the INT4 model and to 15.10%" - it appears that the decline is more steep going to INT8 compared to going to INT4. It would be good to add discussion on this and conduct more experiments to understand this phenomenon and see if there is any saturation.
>
> Thank you for raising this important question. To explore this phenomenon, we conducted additional experiments to extend the analysis across multiple bit-widths (8-bit, 4-bit, 3-bit, and 2-bit) using Llama-2-7b-chat with benign datasets (Ultrachat) for 1 epoch. The results are shown in the table below:
>
> | Llama-2-7b-chat | ASR (8bit/4bit/3bit/2bit) |
> | --------------- | ------------------------- |
> | AQLM            | 17.1 / 18.5 / 28.6 / 40.1 |
> | LLM-QAT         | 15.1 / 16.9 / 25.4 / 36.9 |
> | QLoRA           | 41.7 / 42.3 / 67.3 / 82.0 |
> | AWQ (with ft)   | 10.5 / 17.4 / 29.5 / 38.6 |
> | Q-resafe        | 1.6 / 1.8 / 5.9 / 12.4    |
>
> The results confirm that ASR increases as bit-width decreases across all methods, highlighting that smaller bit-widths reduce the model's capacity to maintain safety. Notably, the steepest ASR growth generally occurs between INT4 and 3-bit, followed by a more gradual increase from 3-bit to 2-bit, suggesting partial saturation at extremely low bit-widths.
>
> The comparison between INT8 and INT4 quantization shows that ASR degradation is more pronounced in INT4, which aligns with the expectation that INT8 retains higher precision, preserving more safety capabilities. Among the methods evaluated, Q-resafe consistently exhibits the lowest ASR and demonstrates minimal deterioration across all bit-widths, owing to its dynamic identification and repair of safety-critical weights. Other methods, such as AWQ and AQLM, show larger ASR increases, with QLoRA experiencing the steepest decline in safety.
>
> We hope this extended analysis addresses your concerns and provides more comprehensive empirical evidence for a better understanding of the relationship between quantization bit-widths and safety.

---

> > ### Comment · Reviewer_Jb1g · 2024-11-25
> > **Thank you!**
> >
> > Thank you for the clarifications and additional analysis.

---

> ### Author Response · Authors · 2024-11-25
> **Thank you for acknowledging our revision plan. We have submitted manuscript with refined safety evaluation claims and enhanced literature review.**
>
> Thank you for your prompt response and for acknowledging our revision plan. We truly appreciate your insightful comments and constructive feedback, especially your suggestion to avoid overly bold claims that could potentially lead to confusion within the community. In light of your valuable input, we have made the necessary revisions and uploaded an updated version of the PDF. This new version addresses the concerns related to the contributions of our safety evaluations, ensuring that the claims are now well-supported and appropriately moderated, as you recommended. Specifically:
>
> > **In the Abstract, we made the following modifications:**
>
> 1. Highlighted that existing safety studies on quantized LLMs already revealed the safety concerns.
> 2. Framed our safety evaluation as a complementary contribution to these existing efforts.
> 3. Substantiated our contribution by specifying that our study covers four mainstream quantization techniques across diverse settings, including varying quantization bit-widths and different quantization-assisting datasets.
>
> > **In Section 1 Introduction, we made the following modifications:**
>
> 1. Removed descriptions suggesting that safety evaluation is underexplored.
> 2. Highlighted that our safety evaluation on quantized LLMs complements existing studies.
> 3. Substantiated the contributions of our safety evaluation by specifying that our study addresses four mainstream quantization techniques across diverse settings, including varying quantization bit-widths and different quantization-assisting datasets.
> 4. We have revised lines 123 to improve clarity and apologize for any confusion caused previously.
>
> > **In Section 2 Related Works, we added a new subsection (Section 2.3) to discuss existing research that pioneered the study on the safety issues of quantized LLMs in detail.**
>
> 1.In line 256, we clarified that this section refers to the ASR values of Llama-2 after adjusting the sampling temperature. Table 2 shows the ASR values of the safety-aligned original model, and Table 3 corresponds to the configuration without fine-tuning.
>
> > **In Section 3, we made the following additions and clarifications:**
>
> 1.We included relevant literature in line 413, providing further context on how our work identifies safety-critical weights.
>
> > **In the Conclusion and Future Work sections, we have discussed the limitations of our work and made adjustments to the formatting details.**
>
> > **Futhermore, we have included new experiment results and clarificaitons requested by all reviewers.**
>
> 1. We conducted an experiment using the Llama-2-7b-chat model with a benign dataset (Ultrachat) for one epoch, extending our safety analysis of quantized models to multiple bit-widths (8/4/3/2 bit-widths).
>
> 2. We conducted another experiment with the Llama-2-7b-chat model and the same benign dataset (Ultrachat) for one epoch, analyzing the impact of identifying safety-critical weights and performing an ablation study.
>
> 3. We incorporated additional widely-used quantization algorithms (LLM.int8(), NF4, and FP4), which are frequently used in practice and could pose significant safety risks to users. Our additional experimental results show that these quantization techniques also compromise the safety of quantized LLMs, and Q-resafe provides a promising solution for mitigating this issue. We believe these results significantly enhance the completeness of our study.
>
> 4. To address concerns about the representativeness and depth of our safety benchmarks, we added experiments using harm-rating metrics as an additional safety assessment. This offers a new perspective on the safety degradation of quantized LLMs, with evaluation cases based on both OpenAI's policy and Llama-2’s policy.
>
> We sincerely hope that these modifications adequately address your concerns and demonstrate the depth of our revision. We are extremely grateful for your constructive suggestions, which have significantly improved the quality and thoroughness of our work. If you have any further suggestions or concerns, please do not hesitate to let us know, and we would be more than happy to address them.
>
> Thank you again for your time and efforts.

---

> ### Author Response · Authors · 2024-12-02
> **Kindly Reminder: Rebuttal Period Ending Soon**
>
> Dear Reviewer Jb1g,
>
> We sincerely appreciate the time and effort you have devoted to reviewing our work. We hope that our responses and revisions have addressed your concerns effectively.  If you are considering revising your score, we would be deeply grateful for your support. Should you have any further questions or suggestions, please do not hesitate to let us know — we would be more than happy to continue the discussion and further improve our work.
>
> Thank you once again for your invaluable feedback and thoughtful engagement.
>
> Best regards,
>
> The author of paper 6792

---

### Note · Authors · 2025-02-16

I have read and agree with the venue's withdrawal policy on behalf of myself and my co-authors.

---

### Meta-Review · Area_Chair_cUfv · 2024-12-20

**Metareview:**

The submission studies the effect of quantization on LLM safety, and how to mitigate it.

+ The paper studies an important but relatively underexplored problem.

- Some of the literature is missing.
- The reviewers found some of the claims to be unsubstantiated.
- The experimental setup needs to be better explained, and the analysis needs to be improved including ablation studies.

**Additional Comments On Reviewer Discussion:**

The paper received borderline ratings, with one reviewer recommending acceptance while the others recommending rejection. The rebuttal was carefully considered by the reviewers, but the issues raised in the initial reviews still remain. While the submission has merit, it is not ready for publication at a top-tier conference.

---

### Decision · Program_Chairs · 2025-01-22

Reject